# HO-SFL: Hybrid-Order Split Federated Learning with Backprop-Free Clients and Dimension-Free Aggregation

**Qiyuan Chen** [1]   **Xian Wu** [1]   **Yi Wang** [1]   **Xianhao Chen** [1]

## Abstract

Fine-tuning large models on edge devices is severely hindered by the memory-intensive back-propagation (BP) in standard frameworks like federated learning and split learning. While substituting BP with zeroth-order optimization can significantly reduce memory footprints, it typically suffers from prohibitively degraded convergence speed. To resolve this dilemma, we propose *Hybrid-Order Split Federated Learning* (HO-SFL). By reformulating the split learning process within a Lagrangian framework, HO-SFL decouples the optimization landscape: The server performs precise first-order updates (i.e., BP), whereas clients conduct memory-efficient zeroth-order optimization. This hybrid design not only eliminates the need for client-side BP but also enables dimension-free model aggregation, drastically lowering communication costs. Crucially, we provide a theoretical convergence analysis, demonstrating that HO-SFL mitigates the dimension-dependent convergence slowdown of zeroth-order optimization, achieving a convergence rate comparable to first-order methods. Extensive experiments on tasks across vision and language modalities validate that HO-SFL achieves convergence speeds comparable to first-order baselines while significantly reducing communication costs and client memory footprints.

## 1. Introduction

Driven by the imperative needs for task-specific-adaptation and data privacy, fine-tuning large models on edge devices—where data is generated—has emerged as a critical demand in the era of artificial intelligence (Malladi et al.,

[1]Department of Electrical and Computer Engineering, The University of Hong Kong, Hong Kong SAR, China. Correspondence to: Xianhao Chen <xcheneee@hku.hk>.

*Proceedings of the 43$^{rd}$ International Conference on Machine Learning*, Seoul, South Korea. PMLR 306, 2026. Copyright 2026 by the author(s).

2023; Peng et al., 2024). Unlike centralized training, which requires aggregating user data, edge learning paradigms enable collaborative model adaptation without compromising user privacy (Lim et al., 2020; Sani et al., 2025).

However, the resource-constrained nature of edge devices poses significant challenges to existing distributed learning frameworks (Lv et al., 2024; Wang et al., 2024). The most prominent framework, federated learning (FL) (McMahan et al., 2017), necessitates that clients perform complete backpropagation (BP) to compute gradients locally. To enable the training of larger models, split federated learning (SFL) (Thapa et al., 2022) was proposed to offload a substantial portion of the model computation to a resource-rich server. Nevertheless, standard SFL and most variants still require clients to execute BP on the client-side sub-model to compute gradients for updates (Oh et al., 2025; Nair et al., 2025). For modern large language models (LLMs) with billions or even trillions of parameters, the memory overhead of storing activations for BP far exceeds the capacity of typical edge hardware, even when the client-side sub-model consists of only a few layers (Lin et al., 2024; Dettmers et al., 2023). Consequently, the memory bottleneck remains a formidable barrier for LLM deployment on edge devices.

As a memory-efficient alternative to BP, zeroth-order (ZO) optimization (Spall, 2002) has garnered increasing attention (Liu et al., 2020). By estimating gradients using only forward passes, ZO optimization eliminates the need to store intermediate activations, drastically reducing memory footprints. Several works have attempted to integrate ZO techniques into FL and SFL to achieve "BP-free" client training (Fang et al., 2022; Li et al., 2025; Liang et al., 2026). However, existing distributed ZO methods often suffer from slow convergence rates compared to first-order methods due to the high variance of gradient estimators and the lack of precise gradient information, particularly in high-dimensional parameter spaces (Shamir, 2017; Ma & Huang, 2025). Therefore, a critical research question arises:

*Can we enable BP-free client training in distributed learning without inheriting the high-dimensional slowdown of zeroth-order optimization?*

We answer this question affirmatively by proposing *Hybrid-*

*Order Split Federated Learning* (HO-SFL). By treating the consistency between client-side outputs and server-side inputs as an equality constraint, we decouple the SFL optimization landscape into two distinct local sub-problems via a Lagrangian framework. In this decoupled regime, the server performs BP to update its parameters and transmits the activation gradients to the clients. These gradients define a local proxy objective for the clients, enabling them to perform memory-efficient ZO updates that align with the global optimization direction. In this way, HO-SFL can leverage the server's computational power for precise first-order updates to maintain high convergence speed, while enabling clients to perform memory-efficient ZO training without the burden of BP. Furthermore, by leveraging shared randomness, HO-SFL achieves dimension-free model aggregation (Li et al., 2025). Specifically, instead of transmitting high-dimensional model parameters, clients need only upload a few scalars to achieve client-side model aggregation, thereby significantly reducing communication overhead compared to other SFL methods.

Our main contributions are summarized as follows:

- We propose HO-SFL, a novel framework that enables BP-free training on clients while maintaining convergence speeds comparable to first-order methods. By reformulating the split learning process within a Lagrangian framework, the server performs first-order updates via BP, while clients leverage the back-transmitted gradients to execute ZO updates. Moreover, HO-SFL enables dimension-free model aggregation, drastically lowering communication costs.

- We theoretically analyze HO-SFL and prove that it achieves a convergence rate of $\mathcal{O}(\sqrt{d_c/PT})$, which effectively mitigates the dimension-dependent convergence slowdown inherent in ZO optimization. Our analysis establishes a theoretical characterization of the dynamics of hybrid-order optimization, providing a rigorous foundation for integrating first-order and zeroth-order optimization methods within a framework.

- We conduct experiments across vision and language modalities to validate our claims. Our results demonstrate that HO-SFL achieves convergence speed comparable to first-order baselines, while reducing client memory usage to inference-only level and minimizing communication cost via dimension-free aggregation.

## 2. Related Work

In this section, we review the literature relevant to our work across three primary dimensions: advances in SFL, the evolution of ZO optimization, and recent works integrating ZO into distributed frameworks.

### 2.1. Advances in Split Federated Learning

Split Federated Learning combines the parallel training of FL with the model splitting of Split Learning (SL) (Gupta & Raskar, 2018; Vepakomma et al., 2018) to reduce client-side computational load. Recent works have focused on optimizing communication efficiency and training latency in SFL (Wu et al., 2023b). SplitFC (Oh et al., 2025) introduces adaptive feature-wise compression, utilizing dropout and quantization strategies based on feature dispersion to reduce communication overhead. To address the high latency caused by the sequential server-side updates in SFL, FSL-SAGE (Nair et al., 2025) employs auxiliary models on clients to estimate server gradients, enabling parallel client training. Furthermore, GAS (Yang & Liu, 2025) introduces an asynchronous SFL framework with generative activation buffers to mitigate the impact of stragglers and biased updates caused by asynchronous transmissions. Unlike these works, which primarily focus on compression or asynchrony while retaining client-side BP, our work focuses on eliminating client-side BP entirely through a hybrid optimization approach.

### 2.2. Zeroth-Order Optimization

ZO optimization estimates gradients using function value differences, avoiding the memory-intensive storage of activation graphs required by BP (Duchi et al., 2015; Nesterov & Spokoiny, 2017). A landmark advancement in this domain is MeZO (Malladi et al., 2023), which adapts ZO-SGD to fine-tune LLMs with memory footprints equivalent to inference. Theoretical advancements have also been made to improve ZO estimators. For instance, Hikima et al. (2025) leverages partial gradient information to shape the covariance matrix of perturbations for more effective update directions, while Ma & Huang (2026) introduces a telescoping series framework to construct unbiased gradient estimators, effectively eliminating the approximation bias inherent in traditional finite-difference methods. While these methods demonstrate the efficacy of ZO in centralized settings, applying them directly to distributed environments without structural adaptation often leads to suboptimal convergence.

### 2.3. Zeroth-Order in Distributed Learning

Recent research has begun to explore the intersection of ZO optimization with distributed learning frameworks (Yi et al., 2022). FedZO (Fang et al., 2022) integrates ZO optimization into FL, enabling clients to update models using stochastic gradient estimators without BP. DeComFL (Li et al., 2025) leverages ZO optimization to achieve dimension-free communication in FL by transmitting random seeds and scalars instead of high-dimensional model parameters. In the context of SFL, MU-SplitFed (Liang et al., 2026) combines ZO optimization with an unbalanced update strategy,

where the server performs multiple update steps for every client step to mitigate straggler effects. However, these methods suffer from the high variance of ZO gradient estimators, resulting in a convergence rate significantly slower than first-order(FO) methods (Ajalloeian, 2019). Hybrid decentralized optimization (Talaei et al., 2025) considers a decentralized network where some nodes use FO gradients and other nodes use ZO estimators. In contrast, HO-SFL hybridizes at the model-split level in SFL, which not only enables BP-free training on clients to minimize memory costs but also achieves a convergence rate comparable to first-order methods.

# 3. Methodology

In this section, we provide the theoretical motivation and the algorithmic design of the proposed *Hybrid-Order Split Federated Learning* (HO-SFL). We reformulate the standard SFL training process through the lens of constrained optimization, which naturally leads to a decoupled learning strategy where the server performs standard BP while the client employs ZO optimization.

## 3.1. System Model

We consider an SFL system with one central server and $M$ clients, indexed by $m \in \{1, \ldots, M\}$. The model is split into a client-side network $f_c(\cdot; \boldsymbol{\theta}_c)$ and a server-side network $f_s(\cdot; \boldsymbol{\theta}_s)$, with parameters $\boldsymbol{\theta} := [\boldsymbol{\theta}_c; \boldsymbol{\theta}_s] \in \mathbb{R}^{d_c + d_s}$. For clarity, we present the system model using a single-sample formulation. We define a data sample as $\xi_m = (\boldsymbol{x}_m, y_m)$ drawn from client $m$'s local distribution $\mathcal{D}_m$. Client $m$ computes the activation $\boldsymbol{z}_m$ via the client-side model $f_c$:

$$\boldsymbol{z}_m = f_c(\boldsymbol{x}_m; \boldsymbol{\theta}_c) \in \mathbb{R}^D. \tag{1}$$

Upon receiving $\boldsymbol{z}_m$, the server completes the forward pass to obtain $\hat{y}_m = f_s(\boldsymbol{z}_m; \boldsymbol{\theta}_s)$ and evaluates the task loss

$$\ell(\boldsymbol{\theta}; \xi_m) := \ell(\hat{y}_m, y_m). \tag{2}$$

For each client $m$, we define the local objective function as

$$\mathcal{L}_m(\boldsymbol{\theta}) := \mathbb{E}_{\xi_m \sim \mathcal{D}_m} \left[ \ell(\boldsymbol{\theta}; \xi_m) \right] \tag{3}$$

and formulate the global optimization problem as

$$\mathcal{L}(\boldsymbol{\theta}) := \frac{1}{M} \sum_{m=1}^{M} \mathcal{L}_m(\boldsymbol{\theta}). \tag{4}$$

It is worth noting that while we use uniform weights here for simplicity, this formulation can be trivially generalized to weighted averaging strategies, such as weighting based on the size of each client's dataset.

## 3.2. Theoretical Motivation: A Lagrangian Perspective

While the goal is to minimize the global loss $\mathcal{L}(\boldsymbol{\theta})$, standard SFL addresses this by directly targeting the nested composite function $\ell(f_s(f_c(\boldsymbol{x}; \boldsymbol{\theta}_c); \boldsymbol{\theta}_s), y)$ as the immediate optimization objective for each data sample $(\boldsymbol{x}, y)$. Unfortunately, this specific per-step formulation creates a *functional coupling* between the client and server parameters. This dependency constrains the system to a unified optimization paradigm, enforcing that both clients and the server simultaneously employ either BP or ZO methods. To decouple the optimization landscapes of the server and client, we use the **variable lifting** (Carreira-Perpinan & Wang, 2014; Wang & Benning, 2023) technique. Specifically, we treat the activations $\boldsymbol{z}$ as an auxiliary variable and reformulate the objective as

$$\begin{aligned} & \ell(f_s(\boldsymbol{z}; \boldsymbol{\theta}_s), y) \\ & \text{s.t.} \quad \boldsymbol{z} = f_c(\boldsymbol{x}; \boldsymbol{\theta}_c). \end{aligned} \tag{5}$$

For this constrained problem, we construct the Lagrangian function $\mathcal{L}_{\lambda}$ by introducing a Lagrange multiplier $\boldsymbol{\lambda}$

$$\mathcal{L}_{\lambda}(\boldsymbol{\theta}_s, \boldsymbol{\theta}_c) = \ell(f_s(\boldsymbol{z}; \boldsymbol{\theta}_s), y) + \boldsymbol{\lambda}^{\top}(f_c(\boldsymbol{x}; \boldsymbol{\theta}_c) - \boldsymbol{z}). \tag{6}$$

This formulation allows us to decompose the global objective into two sub-problems as follows.

**Server-Side Objective ($\mathcal{L}_s$).** Focusing on terms involving $\boldsymbol{\theta}_s$, the server's objective can be rearranged as

$$\mathcal{L}_s(\boldsymbol{\theta}_s) = \ell(f_s(\boldsymbol{z}; \boldsymbol{\theta}_s), y). \tag{7}$$

**Client-Side Objective ($\mathcal{L}_c$).** Focusing on terms involving $\boldsymbol{\theta}_c$, the client's objective is guided by the multiplier, which is given by

$$\mathcal{L}_c(\boldsymbol{\theta}_c) = \boldsymbol{\lambda}^{\top} f_c(\boldsymbol{x}; \boldsymbol{\theta}_c). \tag{8}$$

**Derivation of State Variables.** To solve this, we need to determine the values of $\boldsymbol{z}$ and $\boldsymbol{\lambda}$ at each communication round $t$. First, to enforce consistency, we anchor the auxiliary variable $\boldsymbol{z}$ to the current client forward output

$$\boldsymbol{z}^t = f_c(\boldsymbol{x}; \boldsymbol{\theta}_c^t). \tag{9}$$

Second, the optimal Lagrange multiplier $\boldsymbol{\lambda}^t$ is determined by the stationarity condition $\nabla_{\boldsymbol{z}} \mathcal{L}_{\lambda} = \boldsymbol{0}$ from (6)

$$\nabla_{\boldsymbol{z}} \ell(f_s(\boldsymbol{z}; \boldsymbol{\theta}_s^t), y) - \boldsymbol{\lambda}^t = \boldsymbol{0}. \tag{10}$$

Rearranging the terms and substitute $\boldsymbol{z}$ with $\boldsymbol{z}^t$ yields:

$$\boldsymbol{\lambda}^t = \nabla_{\boldsymbol{z}} \ell(f_s(\boldsymbol{z}^t; \boldsymbol{\theta}_s^t), y). \tag{11}$$

This confirms that the Lagrange multiplier $\boldsymbol{\lambda}^t$ corresponds exactly to the standard gradient of the loss with respect to the input activations.

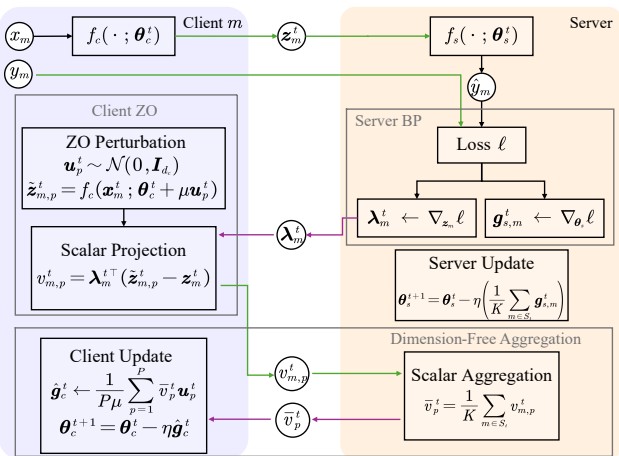

*Figure 1.* **Overview of the HO-SFL training loop.** Each selected client $m$ computes an activation $\boldsymbol{z}_m^t = f_c(\boldsymbol{x}_m; \boldsymbol{\theta}_c^t)$ and sends $(\boldsymbol{z}_m^t, y_m)$ to the server. The server performs BP to update $\boldsymbol{\theta}_s$ and returns the activation-gradient feedback $\boldsymbol{\lambda}_m^t = \nabla_{\boldsymbol{z}_m} \ell$. In parallel, the client runs $P$ ZO perturbation forward passes $\tilde{\boldsymbol{z}}_{m,p}^t = f_c(\boldsymbol{x}_m; \boldsymbol{\theta}_c^t + \mu \boldsymbol{u}_p^t)$ and computes scalar projections $v_{m,p}^t = \boldsymbol{\lambda}_m^{t\top}(\tilde{\boldsymbol{z}}_{m,p}^t - \boldsymbol{z}_m^t)$. The server then aggregates these scalars into $\bar{v}_p^t = \frac{1}{K} \sum_{m \in \mathcal{S}_t} v_{m,p}^t$, which are broadcast back for clients to construct $\hat{\boldsymbol{g}}_c^t = \frac{1}{P\mu} \sum_{p=1}^{P} \bar{v}_p^t \boldsymbol{u}_p^t$ and update $\boldsymbol{\theta}_c$.

### 3.3. The Proposed HO-SFL Framework

Based on the analysis above, we propose HO-SFL. The core training loop and data flow are illustrated in Figure 1, while the detailed procedural steps are summarized in Algorithm 1. Specifically, the server focuses on $\mathcal{L}_s$, updating the server-side parameters $\boldsymbol{\theta}_s$ via BP while simultaneously deriving the Lagrange multiplier $\boldsymbol{\lambda}_m^t$ as consistency feedback. In parallel, each of the $K$ sampled clients leverages the feedback to optimize the client-side objective $\mathcal{L}_c$, through a memory-efficient ZO estimator. Furthermore, by leveraging shared random seeds, clients enable dimension-free aggregation by transmitting only scalars. In the following, we detail the specific steps of HO-SFL during each iteration.

**Client Sampling and Forward.** At the beginning of communication round $t$, the server samples a subset $\mathcal{S}_t \subseteq \{1, \ldots, M\}$ with $|\mathcal{S}_t| = K$. Each sampled client $m$ performs the forward pass on a data sample $\xi_m = (\boldsymbol{x}_m, y_m)$ using its current parameters $\boldsymbol{\theta}_c^t$ to generate $\boldsymbol{z}_m^t$ as defined in Eq. (1). The client transmits $\boldsymbol{z}_m^t$ and label $y_m$ to the server.

**Server-Side Backpropagation.** Upon receiving the data $\{(\boldsymbol{z}_m^t, y_m^t)\}_{m \in \mathcal{S}_t}$, the server completes the forward propagation and computes the loss using Eq. (2). Then, the server performs standard backpropagation to compute

$$\boldsymbol{g}_{s,m}^t = \nabla_{\boldsymbol{\theta}_s} \ell(f_s(\boldsymbol{z}_m^t; \boldsymbol{\theta}_s^t), y_m), \quad \forall m \in \mathcal{S}_t, \quad (12)$$

$$\boldsymbol{\lambda}_m^t = \nabla_{\boldsymbol{z}_m^t} \ell(f_s(\boldsymbol{z}_m^t; \boldsymbol{\theta}_s^t), y_m^t), \quad \forall m \in \mathcal{S}_t. \quad (13)$$

---

**Algorithm 1** Hybrid-Order Split Federated Learning

1: **Initialize:** Model parameters $\boldsymbol{\theta}_c^0$ and $\boldsymbol{\theta}_s^0$; Learning rate $\eta$; Perturbation count $P$; Smoothing parameter $\mu$
2: **Allocate:** Timestamp map $\{t_m'\}_{m=1}^M \leftarrow 0$; History buffers for seeds $\{\{s_p^\tau\}_{p=1}^P\}_\tau$ and scalars $\{\{\bar{v}_p^\tau\}_{p=1}^P\}_\tau$
3: **for** round $t = 0, \ldots, T-1$ **do**
4:     Server samples set $\mathcal{S}_t$; broadcast seeds $\{s_p^t\}_{p=1}^P$
5:     // Phase 1: Synchronization & Forward
6:     **for** each client $m \in \mathcal{S}_t$ **in parallel do**
7:         $t_m' \leftarrow \textbf{\textsc{ClientSync}}(t, t_m')$
8:         Sample $\xi_m = (\boldsymbol{x}_m, y_m)$
9:         Compute activation $\boldsymbol{z}_m^t \leftarrow f_c(\boldsymbol{x}_m; \boldsymbol{\theta}_c^t)$
10:       Client sends $(\boldsymbol{z}_m^t, y_m)$ to Server
11:     **end for**
12:     // Phase 2: Server First-Order Update
13:     **for** each client $m \in \mathcal{S}_t$ **do**
14:         Compute $\hat{y}_m \leftarrow f_s(\boldsymbol{z}_m^t; \boldsymbol{\theta}_s^t)$ and loss $\ell$
15:         Compute gradients $\boldsymbol{g}_{s,m}^t \leftarrow \nabla_{\boldsymbol{\theta}_s} \ell$
16:         Compute feedback $\boldsymbol{\lambda}_m^t \leftarrow \nabla_{\boldsymbol{z}_m^t} \ell$
17:     **end for**
18:     Server update $\boldsymbol{\theta}_s^{t+1} \leftarrow \boldsymbol{\theta}_s^t - \eta \frac{1}{K} \sum_{m \in \mathcal{S}_t} \boldsymbol{g}_{s,m}^t$
19:     Server send $\boldsymbol{\lambda}_m^t$ to client $m$ for $m \in \mathcal{S}_t$
20:     // Phase 3: Client Zeroth-Order Projection
21:     **for** each client $m \in \mathcal{S}_t$ **in parallel do**
22:         $\{v_{m,p}^t\}_{p=1}^P \leftarrow \textbf{\textsc{ZOScalar}}(\boldsymbol{\theta}_c^t, \boldsymbol{\lambda}_m^t, \{s_p^t\}_{p=1}^P)$
23:         Send scalars $\{v_{m,p}^t\}_{p=1}^P$ to Server
24:     **end for**
25:     // Phase 4: Client Aggregation & Update
26:     Server broadcasts $\{\bar{v}_p^t \leftarrow \frac{1}{K} \sum_{m \in \mathcal{S}_t} v_{m,p}^t\}_{p=1}^P$
27:     Clients compute $\hat{\boldsymbol{g}}_c^t \leftarrow \frac{1}{P\mu} \sum_{p=1}^P \bar{v}_p^t \boldsymbol{u}_p^t$
28:     Clients update $\boldsymbol{\theta}_c^{t+1} \leftarrow \boldsymbol{\theta}_c^t - \eta \hat{\boldsymbol{g}}_c^t$
29: **end for**

---

Note that $\boldsymbol{g}_{s,m}^t$ and $\boldsymbol{\lambda}_m^t$ are obtained from a regular BP process, and no extra computing cost is incurred. The server updates its parameter by

$$\boldsymbol{\theta}_s^{t+1} = \boldsymbol{\theta}_s^t - \eta \left( \frac{1}{K} \sum_{m \in \mathcal{S}_t} \boldsymbol{g}_{s,m}^t \right), \quad (14)$$

and transmit gradient $\boldsymbol{\lambda}_m^t$ back to each client.

**Client-Side Zeroth-Order Projection.** To update $\boldsymbol{\theta}_c$ without BP, we employ a ZO estimator to minimize the client-side objective $\mathcal{L}_c$. First, using shared random seeds $\{s_p^t\}_{p=1}^P$, clients generate Gaussian perturbation vectors:

$$\boldsymbol{u}_p^t \sim \mathcal{N}(\boldsymbol{0}, \boldsymbol{I}_{d_c}), \quad p \in \{1, \ldots, P\}. \quad (15)$$

Second, each client computes perturbed activations:

$$\tilde{\boldsymbol{z}}_{m,p}^t = f_c(\boldsymbol{x}_m^t; \boldsymbol{\theta}_c^t + \mu \boldsymbol{u}_p^t), \quad p \in \{1, \ldots, P\}, \quad (16)$$

---

**Algorithm 2** Function: ZO Scalar

---

1: **Function ZOSCALAR**$(\boldsymbol{\theta}_c, \boldsymbol{\lambda}, \{s_p\}_{p=1}^P)$:
2: **for** $p = 1, \ldots, P$ **do**
3:      Generate perturbation $\boldsymbol{u}_p \sim \mathcal{N}(\mathbf{0}, \mathbf{I})$ from seed $s_p$
4:      Perturbed forward: $\tilde{\boldsymbol{z}}_p \leftarrow f_c(\boldsymbol{x}; \boldsymbol{\theta}_c + \mu \boldsymbol{u}_p)$
5:      Compute projection: $v_p \leftarrow \boldsymbol{\lambda}^\top (\tilde{\boldsymbol{z}}_p - \boldsymbol{z})$    ▷ Eq. 17
6: **end for**
7: **Return** Scalars $\{v_p\}_{p=1}^P$

---

**Algorithm 3** Function: Client Synchronization

---

1: **Function CLIENTSYNC**$(t, t')$:
2: **if** $t' < t$ **then**
3:      Fetch history $\{(s_p^\tau, \bar{v}_p^\tau)\}_{p=1}^P$ for $\tau \in \{t', \ldots, t-1\}$
4:      **for** missed round $\tau = t', \ldots, t-1$ **do**
5:          Regenerate $\boldsymbol{u}_p^\tau \leftarrow \text{PRG}(s_p^\tau)$ for $p \in \{1, \ldots, P\}$
6:          Reconstruct grad $\hat{\boldsymbol{g}}_c^\tau \leftarrow \frac{1}{P\mu} \sum_{p=1}^P \bar{v}_p^\tau \boldsymbol{u}_p^\tau$
7:          $\boldsymbol{\theta}_c^{\tau+1} \leftarrow \boldsymbol{\theta}_c^\tau - \eta \hat{\boldsymbol{g}}_c^\tau$    ▷ Sequential Catch-up
8:      **end for**
9: **end if**
10: **Return** $t$    ▷ Update timestamp

---

where $\mu$ is a smoothing parameter. Third, each client computes a scalar projection $v_{m,p}^t$ representing the finite difference of the local objective function $\mathcal{L}_c$. Recall that the local objective is defined as $\mathcal{L}_c(\boldsymbol{\theta}_c) = \boldsymbol{\lambda}^\top f_c(\boldsymbol{x}; \boldsymbol{\theta}_c)$. Setting the unperturbed activation $\boldsymbol{z}_m^t$ as the anchor variable $\boldsymbol{z}$, the difference is calculated as:

$$
\begin{aligned}
v_{m,p}^t &= \mathcal{L}_c(\boldsymbol{\theta}_c^t + \mu \boldsymbol{u}_p^t) - \mathcal{L}_c(\boldsymbol{\theta}_c^t) \\
&= \boldsymbol{\lambda}_m^{t\,\top} (\tilde{\boldsymbol{z}}_{m,p}^t - \boldsymbol{z}_m^t).
\end{aligned}
\tag{17}
$$

This term estimates the directional derivative of the client's output projected onto the loss gradient direction provided by the server. It is important to note that no uplink communication costs are incurred during the above process.

**Global Gradient Reconstruction.** Each sampled client transmits $v_{m,p}^t{}_{p=1}^P$ to the server, which aggregates them across clients:

$$
\bar{v}_p^t = \frac{1}{K} \sum_{m \in \mathcal{S}_t} v_{m,p}^t, \quad p \in \{1, \ldots, P\},
\tag{18}
$$

and broadcasts $\{\bar{v}_p^t\}_{p=1}^P$. Each client constructs:

$$
\hat{\boldsymbol{g}}_c^t = \frac{1}{P\mu} \sum_{p=1}^P \bar{v}_p^t \boldsymbol{u}_p^t,
\tag{19}
$$

followed by the client-side update:

$$
\boldsymbol{\theta}_c^{t+1} = \boldsymbol{\theta}_c^t - \eta \hat{\boldsymbol{g}}_c^t.
\tag{20}
$$

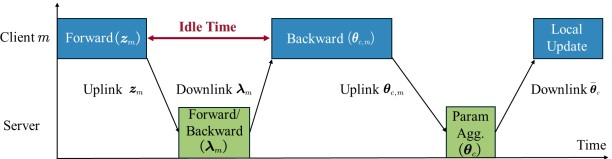

*(a)* Standard SFL Timeline

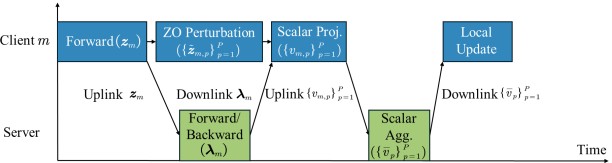

*(b)* HO-SFL Timeline

*Figure 2.* Comparison of system efficiency. (a) Standard SFL incurs significant idle time on the client side waiting for gradients. (b) HO-SFL effectively masks the computational cost of multiple client-side zeroth-order perturbations by overlapping them with the server's backpropagation and communication processes.

It is worth noting that clients only need to upload the scalars $\{v_{m,p}^t\}_{p=1}^P$ for aggregation. In contrast to standard SFL, which mandates the transmission of the full parameter vector scaling with dimension $d_c$, our approach achieves dimension-free uplink communication, thereby significantly reducing the bandwidth cost.

**Client Synchronization Strategy** To accommodate stateless clients that do not participate in every round, HO-SFL incorporates a lightweight synchronization mechanism as outlined in Algorithm 3. Specifically, for a client with a stale model $\boldsymbol{\theta}_c^{t'}$ from round $t' < t$, it requests the server's history of broadcast tuples $\{(s_p^\tau, \bar{v}_p^\tau)\}_{p=1}^P$ corresponding to the missed rounds $\tau \in \{t', \ldots, t-1\}$. The client then iteratively catches up to the current global state by first regenerating the historical perturbation vectors from the random seeds:

$$
\boldsymbol{u}_p^\tau = \text{PRG}(s_p^\tau).
\tag{21}
$$

It then combines them with the scalar projections to reconstruct the gradient estimates:

$$
\hat{\boldsymbol{g}}_c^\tau = \frac{1}{P\mu} \sum_{p=1}^P \bar{v}_p^\tau \boldsymbol{u}_p^\tau.
\tag{22}
$$

These reconstructed gradients are then used to sequentially apply the historical updates:

$$
\boldsymbol{\theta}_c^{\tau+1} \leftarrow \boldsymbol{\theta}_c^\tau - \eta \hat{\boldsymbol{g}}_c^\tau,
\tag{23}
$$

aligning the local model with $\boldsymbol{\theta}_c^t$ without the need to download high-dimensional model parameters.

### 3.4. Discussions: Efficiency of HO-SFL

While the primary motivation of HO-SFL is to overcome the memory barrier of clients, we also highlight two *ex-*

*tra* advantages of HO-SFL concerning communication and latency efficiency (as illustrated in Figure 2).

**Dimension-Free Aggregation.** Regarding client-side aggregation, HO-SFL significantly alleviates the communication bottleneck. In standard SFL, clients transmit high-dimensional model parameters whose size scales linearly with the client-side model dimension $d_c$, resulting in a communication complexity of $\mathcal{O}(d_c)$. In contrast, HO-SFL enables dimension-free aggregation by requiring each client to transmit only $P$ scalars $\{v_{m,p}\}_{p=1}^{P}$ per communication round, where $P$ is independent of the model architecture and typically satisfies $P \ll d_c$. Thus, the communication cost for model aggregation is reduced to $\mathcal{O}(P)$.

**Latency Hiding via Decoupling.** In standard SFL, clients must remain idle while waiting for the activation gradient from the server. Benefiting from the decoupling strategy, HO-SFL can effectively overlap both computation and communication: the client-side perturbation can be overlapped with server-side forward/backward as well as the activation uplink and gradient downlink processes, thereby potentially shortening end-to-end latency. We conducted a simulation analysis based on the LLaMA-3.2-1B (Grattafiori et al., 2024) under realistic edge computing conditions and validated the feasibility of latency hiding in Appendix C.1.

## 4. Convergence Analysis

In this section, we establish the theoretical guarantees for the proposed HO-SFL framework. Unlike standard SFL, HO-SFL involves a hybrid optimization landscape where the server performs first-order updates while clients perform ZO updates. We analyze the convergence properties of HO-SFL under non-convex settings and quantify the impact of the zeroth-order approximation error on the global convergence rate. Detailed proofs are provided in Appendix A.

### 4.1. Assumptions

We adopt standard assumptions widely used in the analysis of non-convex federated optimization (Han et al., 2024).

**Assumption 4.1** (Smoothness). *The global objective $\mathcal{L}(\boldsymbol{\theta})$ is $\beta$-smooth, i.e., for any $\boldsymbol{\theta}_1, \boldsymbol{\theta}_2$,*

$$\|\nabla\mathcal{L}(\boldsymbol{\theta}_1) - \nabla\mathcal{L}(\boldsymbol{\theta}_2)\| \leq \beta\|\boldsymbol{\theta}_1 - \boldsymbol{\theta}_2\|.$$

**Assumption 4.2** (Unbiased Stochastic Gradients, Bounded Variance, and Heterogeneity). *Client $m$ draws an independent data sample $\xi_m \sim \mathcal{D}_m$ and forms a stochastic gradient $\boldsymbol{g}_m := \nabla_{\boldsymbol{\theta}}\ell(\boldsymbol{\theta}; \xi_m)$. We assume*

1. **Unbiasedness:** $\mathbb{E}_{\xi_m}[\boldsymbol{g}_m] = \nabla\mathcal{L}_m(\boldsymbol{\theta})$.

2. **Bounded Variance:** $\mathbb{E}_{\xi_m}\left[\|\boldsymbol{g}_m - \nabla\mathcal{L}_m(\boldsymbol{\theta})\|^2\right] \leq \sigma^2$.

3. **Bounded Gradient Dissimilarity:** $\|\nabla\mathcal{L}_m(\boldsymbol{\theta}) - \nabla\mathcal{L}(\boldsymbol{\theta})\|^2 \leq \kappa^2$.

### 4.2. Properties of the Hybrid Estimator

A core challenge in analyzing HO-SFL is the bias and variance introduced by the zeroth-order gradient estimator on the client side. Based on the detailed derivations in **Lemma A.2** and **Lemma A.5**, we characterize the behavior of the client-side estimator $\hat{\boldsymbol{g}}_c^t$.

**Proposition 4.3** (Bias and Variance Decomposition). *Let $\Gamma$ be the regularity bound on the gradient magnitudes and Hessian spectral norms. The client-side zeroth-order estimator $\hat{\boldsymbol{g}}_c^t$ exhibits the following properties:*

*1. Bias Control (from Lemma A.6): The estimator is biased due to the curvature of the loss landscape. The expected deviation is bounded by the smoothing parameter $\mu$ and the client model dimension $d_c$:*

$$\|\mathbb{E}_t[\hat{\boldsymbol{g}}_c^t] - \nabla_{\boldsymbol{\theta}_c}\mathcal{L}(\boldsymbol{\theta}^t)\|^2 \leq \frac{\mu^2\Gamma^4}{4}(d_c + 3)^3. \quad (24)$$

*2. Second Moment Bound (from Lemma A.4): The second moment of the aggregated client estimator $\hat{\boldsymbol{g}}_c^t$ is bounded by the true gradient norm scaled by an expansion factor $C_1$, plus a composite variance term:*

$$\mathbb{E}_t\|\hat{\boldsymbol{g}}_c^t\|^2 \leq C_1\|\nabla_{\boldsymbol{\theta}_c}\mathcal{L}(\boldsymbol{\theta}^t)\|^2 + \Omega_c, \quad (25)$$

*where $\Omega_c := C_1(\sigma^2 + \kappa^2) + \sigma_{ZO}^2$ collects the combined client-side variance, including stochastic gradient noise ($\sigma^2$), client heterogeneity ($\kappa^2$), and zeroth-order estimation error ($\sigma_{ZO}^2$).*

### 4.3. Main Convergence Result

Building on the properties of the hybrid estimator, we now establish the convergence guarantees of HO-SFL under the non-convex setting.

**Theorem 4.4** (Convergence Bound of HO-SFL). *Suppose Assumptions 4.1 and 4.2 hold. Let the learning rate satisfy $\eta \leq \frac{1}{2\beta C_1}$. For any $T \geq 1$, the average squared gradient norm of HO-SFL is bounded by:*

$$\frac{1}{T}\sum_{t=0}^{T-1}\mathbb{E}\|\nabla\mathcal{L}(\boldsymbol{\theta}^t)\|^2 \leq \frac{4\Delta_{\mathcal{L}}}{\eta T} + 2\eta\beta\left(\Omega_c + \Omega_s\right)$$
$$+ \frac{\mu^2\Gamma^4}{2}(d_c + 3)^3, \quad (26)$$

*where $\Delta_{\mathcal{L}} = \mathcal{L}(\boldsymbol{\theta}^0) - \mathcal{L}^*$, $\Omega_s = \frac{\sigma^2 + \kappa^2}{K}$ represents the server-side variance, and $\Omega_c = C_1(\sigma^2 + \kappa^2) + \sigma_{ZO}^2$ denotes the aggregated client-side variance.*

Theorem 4.4 provides a finite-time error bound on the gradient norm. The bound consists of the optimization error, the

variance term and the bias term induced by the ZO approximation. To explicitly quantify the convergence rate with respect to the dimension $d_c$ and the perturbation number $P$, we present the following corollary.

**Corollary 4.5** (Convergence Rate of HO-SFL)**.** *By choosing the learning rate* $\eta = \Theta\left(\sqrt{\frac{P}{T d_c}}\right)$ *and the smoothing parameter* $\mu = \mathcal{O}\left((PT)^{-1/4} d_c^{-5/4}\right)$*, the convergence rate of HO-SFL satisfies:*

$$\min_{0 \le t < T} \mathbb{E}\|\nabla\mathcal{L}(\boldsymbol{\theta}^t)\|^2 = \mathcal{O}\left(\sqrt{\frac{d_c}{PT}}\right). \qquad (27)$$

*Proof.* Substituting the selected $\eta$ into the bound in Eq. (26), the optimization error term $\mathcal{O}(\frac{1}{\eta T})$ and the variance term $\mathcal{O}(\eta \frac{d_c}{P})$ are balanced to scale as $\mathcal{O}(\sqrt{\frac{d_c}{PT}})$. Simultaneously, the chosen scaling for $\mu$ ensures that the bias term $\mathcal{O}(\mu^2 d_c^3)$ does not exceed the order of the dominant term. Thus, the total convergence rate is dominated by $\mathcal{O}(\sqrt{\frac{d_c}{PT}})$. $\qquad\square$

*Remark* 4.6. Standard ZO optimization methods typically suffer from a linear dependence on the problem dimension, yielding a rate of $\mathcal{O}(\sqrt{d/T})$ (where $d$ is the full model dimension). In contrast, Corollary 4.5 highlights the critical advantage of **dimensionality decoupling** in HO-SFL. The dimension factor in the convergence rate depends only on the client-side dimension $d_c$, where $d_c \ll d$ in split learning settings. This effectively isolates the ZO optimization difficulty to a much smaller subspace, significantly mitigating the variance of the gradient estimator, which would otherwise be exacerbated by the massive server-side parameters.

## 5. Experiments

In this section, we evaluate the performance of HO-SFL against state-of-the-art SFL baselines on vision and language tasks to demonstrate its fast convergence speed, communication efficiency, and substantial reductions in client memory usage.

### 5.1. Experimental Setup

**Datasets and Models.** For vision tasks, we utilize CIFAR-10 and CIFAR-100 datasets (Krizhevsky et al., 2009). We evaluate under both IID and Non-IID settings, where Non-IID partitions are generated using a Dirichlet distribution with $\alpha = 1$ (Hsu et al., 2019). The model is a pre-trained ResNet-18 (He et al., 2016) and split between the 2nd and 3rd residual blocks. For language tasks, we employ five pre-trained causal language models of varying scales: OPT-125M (Zhang et al., 2022), Gemma-3-270M (Team et al., 2025), LLaMA-3.2-1B, LLaMA-3.2-3B (Grattafiori et al., 2024), and Qwen3-8B (Yang et al., 2025). These models are

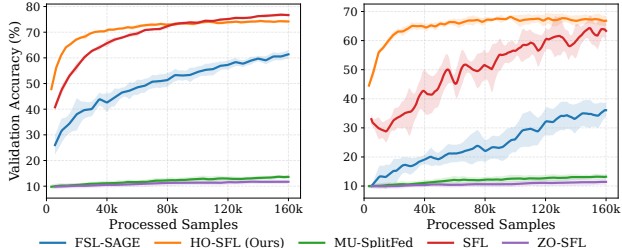

*Figure 3.* Validation accuracy convergence on CIFAR-10 under IID (left) and Non-IID (right) settings. Solid lines denote the mean performance, and shaded regions represent the standard deviation across 10 independent random seeds.

evaluated on a subset of the GLUE benchmark (Wang et al., 2018) (SST-2, RTE, WSC) and SQuAD (Rajpurkar et al., 2016) using LoRA (Hu et al., 2022). We use a larger-scale partial-participation setup for vision tasks (100 total clients, 10 sampled per round) and a smaller-scale setup for LLM tasks (10 total clients, 3 sampled per round).

**Baselines and Fairness Criterion.** We benchmark HO-SFL against multiple methods categorized by their optimization strategy. The first-order baselines include **SFL** (Thapa et al., 2022), **SplitLoRA** (Lin et al., 2024), and **FSL-SAGE** (Nair et al., 2025), while the zeroth-order baselines comprise **ZO-SFL** and **MU-SplitFed** (Liang et al., 2026). Unless otherwise specified, we set the number of perturbations $P$ in HO-SFL to 5 for vision tasks and 2 for language tasks, and uniformly set $\mu$ to $1 \times 10^{-3}$. Crucially, since HO-SFL and MU-SplitFed perform aggregation at every step, whereas other methods synchronize after multiple local updates, comparisons based on communication rounds are inherently misleading. To ensure a rigorous evaluation, we adopt *proceeded samples* as the unified metric, terminating training when the aggregate number of processed samples reaches 160k for vision tasks and 80k for language tasks. Detailed hyperparameter configurations, specific baseline introduction, and a comprehensive discussion on the fairness of the metric are provided in Appendix B.

### 5.2. Performance Analysis

**Vision Tasks.** Figure 3 visualizes the validation accuracy against processed samples on CIFAR-10. In the **IID setting**, HO-SFL exhibits only a marginal performance gap compared to SFL, demonstrating that HO-SFL effectively mitigates the convergence slowdown associated with ZO estimation. However, in **Non-IID setting**, HO-SFL demonstrates superior performance, even outperforming the first-order baseline (i.e., SFL). Unlike other methods that suffer from severe client drift due to infrequent aggregation of high-dimensional models, HO-SFL executes dimension-free aggregation at every step, effectively mimicking centralized training. Furthermore, pure zeroth-order approaches like

MU-SplitFed and ZO-SFL struggle to converge within the limited sample budget, highlighting the efficiency of our proposed method. Due to space constraints, the CIFAR-100 convergence curves and the test accuracies with standard deviations are provided in Appendix C.2

**Language Tasks.** Table 1 presents the fine-tuning results on GLUE tasks, demonstrating that HO-SFL maintains highly competitive performance against SplitLoRA (the first-order baseline) despite relying on zeroth-order client-side updates. Across varying model architectures, our method exhibits remarkable robustness; for instance, HO-SFL achieves a comparable 93.2% accuracy on SST2 and notably surpasses SplitLoRA on the RTE task when fine-tuning the LLaMA-3.2-1B. In contrast, the standard ZO-SFL baseline fails to converge effectively on most tasks. To further assess the scalability of HO-SFL and its performance on generative tasks, we extend our evaluation to the SQuAD dataset. As shown in Table 2, we scale the model size by a factor of $64\times$, ranging from the 125M lightweight model to the 8B foundation model. Across this scaling range, HO-SFL exhibits no convergence degradation and matches the F1 score of the SplitLoRA. These results demonstrate that HO-SFL scales robustly to large LLMs and validate the effectiveness of our hybrid formulation in stabilizing convergence.

*Table 1.* LLM Fine-tuning Accuracy (%) on GLUE tasks. Values in boldface indicate the highest accuracy.

| Model | Task | SplitLoRA | ZO-SFL | **HO-SFL** |
|---|---|---|---|---|
| OPT (125M) | SST2 | 87.5 | 52.8 | **87.6** |
| | WSC | **64.4** | 36.5 | 60.6 |
| | RTE | 57.8 | 52.0 | **59.2** |
| Gemma-3 (270M) | SST2 | 90.3 | 51.8 | **90.8** |
| | WSC | 61.5 | 36.5 | **62.5** |
| | RTE | 59.6 | 54.2 | **65.0** |
| LLaMA-3.2 (1B) | SST2 | **94.4** | 61.5 | 93.9 |
| | WSC | 61.5 | 36.5 | **61.5** |
| | RTE | 70.0 | 49.1 | **73.3** |

*Table 2.* LLM Fine-tuning F1 Scores on the SQuAD task.

| Model | SplitLoRA | **HO-SFL** |
|---|---|---|
| OPT (125M) | 0.5985 | 0.5744 |
| LLaMA-3.2 (1B) | 0.8804 | 0.8687 |
| LLaMA-3.2 (3B) | 0.9271 | 0.9238 |
| Qwen3 (8B) | 0.9413 | 0.9389 |

### 5.3. Communication and Memory Efficiency

**Communication Analysis.** We analyze the communication cost breakdown in Figure 4a recorded in the CIFAR-10 IID experiment. Overall, HO-SFL achieves the lowest commu-

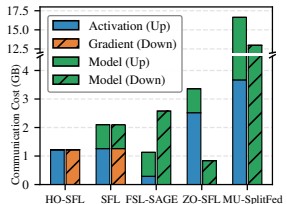
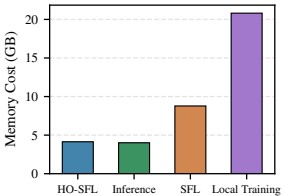

*(a)* Breakdown of communication traffic.

*(b)* Client-side peak memory consumption

*Figure 4.* Communication and memory profiling.

nication cost among the compared methods (2.44 GB), substantially lower than SFL (4.20 GB) and ZO-SFL (4.20 GB), and also lower than FSL-SAGE (3.71 GB). The savings mainly come from avoiding high-dimensional model transmission during aggregation, whereas MU-SplitFed incurs a massive overhead due to transmitting full model updates at every step. For readability, Figure 4a omits the scalar/seed components (e.g., HO-SFL seeds/scalars and the scalar traffic in ZO-SFL and Mu-SplitFed), each contributing less than 1 MB in total; see Appendix C.5 for the exact numbers.

**Memory Consumption.** We profiled the peak client-side memory usage for fine-tuning LLaMA-3.2-1B with LoRA on the SST-2 task, as illustrated in Figure 4b. In this profiling setup, the client-side split contains 8 transformer layers. While full local training demands a prohibitive 20.81 GB that far exceeds the capacity of typical edge devices, even standard SFL imposes significant pressure by consuming 8.78 GB. In contrast, HO-SFL dramatically reduces this footprint to 4.14 GB. This is strikingly close to inference( 4.01GB), confirming that HO-SFL effectively eliminates the memory bottleneck of BP. Detailed information regarding the memory measurement methodology and device specifications is provided in Appendix C.6.

### 5.4. Ablation Studies

Due to space constraints, we present detailed ablation studies in Appendix C.3 and Appendix C.4. Experiments on LLaMA-3.2-1B varying the client-side transformer layers from 2 to 8 confirm our theoretical analysis in Corollary 4.5. While increasing the client-side dimension $d_c$ leads to a marginal reduction in convergence speed due to the dimension-dependent variance of the ZO estimator, HO-SFL exhibits strong robustness, effectively mitigating the severe slowdown typically observed in pure ZO methods. Results on ResNet-18 confirm that increasing the perturbation count $P$ yields monotonic performance improvements. Meanwhile, the smoothing parameter $\mu$ requires a trade-off: values that are too large increase the approximation bias (Proposition 4.3), while values that are too small suffer from numerical precision issues (Jongeneel et al., 2024).

# 6. Discussions

Beyond the theoretical and empirical results above, we further discuss practical considerations for deploying HO-SFL, including its relation to memory-efficient techniques, the runtime trade-offs regarding hyperparameter selection, and the privacy considerations inherent to our framework.

**Alternative Memory-Efficient Techniques.** Standard memory-efficient techniques can alleviate resource constraints on edge devices, but most of them still fundamentally rely on client-side BP. For instance, activation checkpointing (Chen et al., 2016) reduces activation storage through recomputation, but its memory footprint remains above the inference level and the additional forward passes introduce extra latency. Similarly, shallow partitioning, which assigns fewer layers to the client, reduces the number of client-side parameters but does not remove the need to store activations for BP; thus, the client memory footprint can still grow rapidly with sequence length. In contrast, HO-SFL eliminates client-side BP and maintains an inference-level memory footprint. Importantly, architectural strategies such as LoRA (Hu et al., 2022) and shallow partitioning are orthogonal to HO-SFL and can be integrated to further improve memory and communication efficiency.

**Runtime Trade-offs and Hyperparameter Selection.** A key design choice in HO-SFL is selecting the number of perturbations $P$, which determines the trade-off between gradient estimation accuracy and runtime latency. As shown in Figure 8a of Appendix C.4, increasing $P$ consistently improves convergence, while the marginal gains gradually diminish. Importantly, due to the latency-hiding mechanism discussed in Subsection 3.4, the client-side ZO computations incur zero additional latency as long as they finish within the server's BP and communication idle time. Therefore, we recommend a principled selection of $P$: maximizing it up to the threshold where the runtime curve begins to bend upwards, ensuring optimal gradient estimation without sacrificing training speed.

**Privacy and Security Considerations.** We acknowledge the risk that transmitting intermediate activations from clients to the server could potentially expose local data to feature-space inversion attacks. However, it is crucial to note that transmitting activations is the fundamental mechanism of *all* SFL frameworks; this vulnerability is inherently inherited from the standard SFL architecture rather than introduced by HO-SFL. Because our contribution fundamentally alters the optimization landscape rather than the underlying data flow, HO-SFL remains compatible with standard privacy-enhancing defenses. For instance, techniques such as applying Differential Privacy noise (Wu et al., 2023a) to the activations can be seamlessly integrated into HO-SFL to

provide rigorous privacy guarantees without hindering our hybrid-order optimization.

# 7. Conclusion

In this paper, we propose Hybrid-Order Split Federated Learning, a novel framework that resolves the tension between memory constraints and convergence speed when fine-tuning large models on edge devices. By reformulating the split learning process through a Lagrangian perspective, HO-SFL decouples optimization into server-side first-order and client-side zeroth-order updates. Our theoretical analysis shows it mitigates the dimension-dependent convergence slowdown inherent to pure zeroth-order methods. Empirically, HO-SFL achieves convergence performance comparable to fully first-order baselines, while reducing client memory consumption to inference levels and minimizing communication overhead via dimension-free aggregation.

# Acknowledgements

This work was supported in part by the Research Grants Council of Hong Kong under Grant 27213824 and CRS HKU702/2.

# Impact Statement

This paper presents work whose goal is to advance the field of Machine Learning. There are many potential societal consequences of our work, none of which we feel must be specifically highlighted here

# Code Availability

Our implementation of Hybrid-Order Split Federated Learning (HO-SFL) is available at https://github.com/HKU-WILL-Lab/HO-SFL.

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

# A. Proofs

## A.1. Notions

Let $\mathcal{F}_t$ be the filtration generated by all randomness up to the end of round $t - 1$, i.e.,

$$\mathcal{F}_t := \sigma\big(\boldsymbol{\theta}^0, \{(\mathcal{S}_\tau, \boldsymbol{\xi}^\tau, \mathcal{U}^\tau)\}_{\tau=0}^{t-1}\big).$$

In particular, $\boldsymbol{\theta}^t$ is $\mathcal{F}_t$-measurable. We use $\mathbb{E}_t[\cdot] := \mathbb{E}[\cdot \mid \mathcal{F}_t]$ to denote conditional expectation given the history. At round $t$, the algorithm involves three mutually independent sources of randomness: the subset of sampled clients $\mathcal{S}_t$, the data samples $\boldsymbol{\xi}^t := \{\xi_m^t\}_{m \in \mathcal{S}_t}$ drawn by the selected clients, and the random perturbation vectors $\mathcal{U}^t := \{\boldsymbol{u}_p^t\}_{p=1}^P$. To maintain notational conciseness, we define any expectation with a subscript of round $t$ variables to implicitly condition on $\mathcal{F}_t$. For instance, $\mathbb{E}_{\boldsymbol{\xi}^t}[\cdot] := \mathbb{E}_{\boldsymbol{\xi}^t}[\cdot \mid \mathcal{F}_t]$ and $\mathbb{E}_{\mathcal{S}_t}[\cdot] := \mathbb{E}_{\mathcal{S}_t}[\cdot \mid \mathcal{F}_t]$.

For each selected client $m \in \mathcal{S}_t$, the forward pass produces smashed data $\boldsymbol{z}_m^t := f_c(x_m^t; \boldsymbol{\theta}_c^t) \in \mathbb{R}^D$ and perturbed smashed data $\tilde{\boldsymbol{z}}_{m,p}^t := f_c(x_m^t; \boldsymbol{\theta}_c^t + \mu \boldsymbol{u}_p^t)$. The server returns the activation gradient (dual variable) $\boldsymbol{\lambda}_m^t := \nabla_{\boldsymbol{z}_m^t} \ell(\boldsymbol{\theta}^t; \xi_m^t) \in \mathbb{R}^D$. We also denote the local stochastic gradients at round $t$ as

$$\boldsymbol{g}_{c,m}^t := \nabla_{\boldsymbol{\theta}_c} \ell(\boldsymbol{\theta}^t; \xi_m^t), \qquad \boldsymbol{g}_{s,m}^t := \nabla_{\boldsymbol{\theta}_s} \ell(\boldsymbol{\theta}^t; \xi_m^t), \tag{28}$$

so $\boldsymbol{\lambda}_m^t$, $\boldsymbol{g}_{c,m}^t$, and $\boldsymbol{g}_{s,m}^t$ are measurable with respect to $\sigma(\mathcal{F}_t, \mathcal{S}_t, \xi_m^t)$. We define their aggregated versions as

$$\hat{\boldsymbol{g}}_c^t := \frac{1}{K} \sum_{m \in \mathcal{S}_t} \hat{\boldsymbol{g}}_{c,m}^t, \qquad \boldsymbol{g}_s^t := \frac{1}{K} \sum_{m \in \mathcal{S}_t} \boldsymbol{g}_{s,m}^t. \tag{29}$$

We denote the aggregated gradients at round $t$ as $\boldsymbol{g}_s^t$ (server-side) and $\hat{\boldsymbol{g}}_c^t$ (client-side), such that the global update rule is $\boldsymbol{\theta}^{t+1} = \boldsymbol{\theta}^t - \eta[\hat{\boldsymbol{g}}_c^t; \boldsymbol{g}_s^t]$.

For brevity, we denote the full parameter vector by $\boldsymbol{\theta} := [\boldsymbol{\theta}_c; \boldsymbol{\theta}_s]$. The global objective function is given by:

$$\mathcal{L}(\boldsymbol{\theta}) := \frac{1}{M} \sum_{m=1}^M \mathcal{L}_m(\boldsymbol{\theta}), \qquad \mathcal{L}_m(\boldsymbol{\theta}) := \mathbb{E}_{\xi_m \sim \mathcal{D}_m}[\ell(\boldsymbol{\theta}; \xi_m)], \tag{30}$$

where $\ell(\boldsymbol{\theta}; \xi_m)$ is the sample-wise loss for a single data sample $\xi_m = (x_m, y_m)$ drawn from the local distribution $\mathcal{D}_m$.

To facilitate the analysis of the zeroth-order estimator, we formally define the regularity bound $\Gamma$, which encapsulates the upper bounds on the dual variables and the derivatives of the client network.

**Definition A.1** (Regularity Bound $\Gamma$). Let $\mathcal{H}_{m,p}^t := \nabla_{\boldsymbol{\theta}_c}^2 f_c(x_m^t; \boldsymbol{\theta}_c^t + \tau_{m,p}^t \mu \boldsymbol{u}_p^t)$ denote the Hessian of the client model evaluated at the intermediate point determined by Taylor's theorem (where $\tau_{m,p}^t \in (0,1)$). We define the regularity bound $\Gamma_t$ at round $t$ as:

$$\Gamma_t := \max\left\{ \max_{m \in \mathcal{S}_t} \|\boldsymbol{\lambda}_m^t\|, \ \max_{m \in \mathcal{S}_t} \|\nabla_{\boldsymbol{\theta}_c} f_c(x_m^t; \boldsymbol{\theta}_c^t)\|_{\mathrm{op}}, \ \max_{m \in \mathcal{S}_t, \, p \in \{1,\dots,P\}} \|\mathcal{H}_{m,p}^t\|_{\mathrm{op}} \right\}, \tag{31}$$

where $\|\cdot\|_{\mathrm{op}}$ denotes the operator norm. We further define the global bound $\Gamma := \sup_{0 \le t \le T-1} \Gamma_t$.

## A.2. Assumptions

We adopt standard assumptions widely used in the analysis of non-convex federated optimization and zeroth-order methods.

**Assumption 4.1** (Smoothness). The global objective $\mathcal{L}(\boldsymbol{\theta})$ is $\beta$-smooth, i.e., for any $\boldsymbol{\theta}_1, \boldsymbol{\theta}_2$,

$$\|\nabla\mathcal{L}(\boldsymbol{\theta}_1) - \nabla\mathcal{L}(\boldsymbol{\theta}_2)\| \le \beta\|\boldsymbol{\theta}_1 - \boldsymbol{\theta}_2\|.$$

**Assumption 4.2** (Unbiased Stochastic Gradients, Bounded Variance, and Heterogeneity). Client $m$ draws an independent data sample $\xi_m \sim \mathcal{D}_m$ and forms a stochastic gradient $\boldsymbol{g}_m := \nabla_{\boldsymbol{\theta}} \ell(\boldsymbol{\theta}; \xi_m)$. We assume

1. **Unbiasedness:** $\mathbb{E}_{\xi_m}[\boldsymbol{g}_m] = \nabla\mathcal{L}_m(\boldsymbol{\theta})$.

2. **Bounded Variance:** $\mathbb{E}_{\xi_m}\big[\|\boldsymbol{g}_m - \nabla\mathcal{L}_m(\boldsymbol{\theta})\|^2\big] \le \sigma^2$.

3. **Bounded Gradient Dissimilarity:** $\|\nabla\mathcal{L}_m(\boldsymbol{\theta}) - \nabla\mathcal{L}(\boldsymbol{\theta})\|^2 \le \kappa^2$.

## A.3. Proof of Lemma A.2: Bound on Local ZO Estimator

**Lemma A.2** (Conditional Bound on Local ZO Estimator). *Condition on $(\mathcal{F}_t, \mathcal{S}_t, \xi_m^t)$, so that $\boldsymbol{\theta}_c^t$, $\boldsymbol{\lambda}_m^t$, and $\boldsymbol{g}_{c,m}^t$ are fixed. Let*

$$\hat{\boldsymbol{g}}_{c,m}^t = \frac{1}{P\mu} \sum_{p=1}^{P} v_{m,p}^t \boldsymbol{u}_p^t, \qquad \boldsymbol{u}_p^t \sim \mathcal{N}(\boldsymbol{0}, \boldsymbol{I}_{d_c}) \text{ i.i.d.} \tag{32}$$

*where*

$$v_{m,p}^t := (\boldsymbol{\lambda}_m^t)^\top (\tilde{\boldsymbol{z}}_{m,p}^t - \boldsymbol{z}_m^t), \qquad \tilde{\boldsymbol{z}}_{m,p}^t := f_c(x_m^t; \boldsymbol{\theta}_c^t + \mu \boldsymbol{u}_p^t), \; \boldsymbol{z}_m^t := f_c(x_m^t; \boldsymbol{\theta}_c^t). \tag{33}$$

*Let $\boldsymbol{g}_{c,m}^t := \nabla_{\boldsymbol{\theta}_c} \ell(\boldsymbol{\theta}^t; \xi_m^t)$. Using the regularity bound $\Gamma$ from Definition A.1, we have:*

$$\mathbb{E}_{\mathcal{U}^t}\left[ \left\| \hat{\boldsymbol{g}}_{c,m}^t \right\|^2 \right] \leq C_1 \left\| \boldsymbol{g}_{c,m}^t \right\|^2 + \sigma_{ZO}^2, \tag{34}$$

*with*

$$C_1 := 2\left( 1 + \frac{d_c + 1}{P} \right), \qquad \sigma_{ZO}^2 := \frac{\mu^2}{2} d_c(d_c + 2)(d_c + 4) \Gamma^4. \tag{35}$$

*Proof.* By Taylor's theorem with Lagrange remainder, there exists some $\tau_{m,p}^t \in (0,1)$ such that:

$$\begin{aligned} \Delta \boldsymbol{z}_{m,p}^t &= \tilde{\boldsymbol{z}}_{m,p}^t - \boldsymbol{z}_m^t \\ &= f_c(x_m^t; \boldsymbol{\theta}_c^t + \mu \boldsymbol{u}_p^t) - f_c(x_m^t; \boldsymbol{\theta}_c^t) \\ &\overset{(a)}{=} \mu \, \mathcal{J}_m^t \, \boldsymbol{u}_p^t + \underbrace{\frac{\mu^2}{2} \, \mathcal{H}_{m,p}^t [\boldsymbol{u}_p^t, \boldsymbol{u}_p^t]}_{=:\boldsymbol{R}_{m,p}^t}. \end{aligned} \tag{36}$$

Here $\mathcal{J}_m^t := \nabla_{\boldsymbol{\theta}_c} f_c(x_m^t; \boldsymbol{\theta}_c^t) \in \mathbb{R}^{D \times d_c}$ is the Jacobian matrix, and $\mathcal{H}_{m,p}^t := \nabla_{\boldsymbol{\theta}_c}^2 f_c(x_m^t; \boldsymbol{\theta}_c^t + \tau_{m,p}^t \mu \boldsymbol{u}_p^t)$ is the Hessian tensor. $\mathcal{H}_{m,p}^t[\cdot, \cdot]$ denotes its bilinear action on the perturbation vectors. Using (36) in the definition of $v_{m,p}^t$ gives

$$\begin{aligned} v_{m,p}^t &= (\boldsymbol{\lambda}_m^t)^\top \Delta \boldsymbol{z}_{m,p}^t \\ &= \mu (\boldsymbol{\lambda}_m^t)^\top \mathcal{J}_m^t \boldsymbol{u}_p^t + (\boldsymbol{\lambda}_m^t)^\top \boldsymbol{R}_{m,p}^t \\ &\overset{(b)}{=} \mu (\boldsymbol{u}_p^t)^\top \boldsymbol{g}_{c,m}^t + (\boldsymbol{\lambda}_m^t)^\top \boldsymbol{R}_{m,p}^t, \end{aligned} \tag{37}$$

where $(b)$ follows from the chain rule:

$$\boldsymbol{g}_{c,m}^t = \nabla_{\boldsymbol{\theta}_c} \ell(\boldsymbol{\theta}^t; \xi_m^t) = (\nabla_{\boldsymbol{\theta}_c} f_c(x_m^t; \boldsymbol{\theta}_c^t))^\top \nabla_{\boldsymbol{z}_m^t} \ell(\boldsymbol{\theta}^t; \xi_m^t) = (\mathcal{J}_m^t)^\top \boldsymbol{\lambda}_m^t. \tag{38}$$

Substituting (37) into the definition of the gradient estimator $\hat{\boldsymbol{g}}_{c,m}^t$ yields two distinct terms:

$$\begin{aligned} \hat{\boldsymbol{g}}_{c,m}^t &= \frac{1}{P\mu} \sum_{p=1}^{P} v_{m,p}^t \boldsymbol{u}_p^t \\ &= \underbrace{\frac{1}{P} \sum_{p=1}^{P} \boldsymbol{u}_p^t (\boldsymbol{u}_p^t)^\top \boldsymbol{g}_{c,m}^t}_{=:\, \boldsymbol{A}} + \underbrace{\frac{1}{P\mu} \sum_{p=1}^{P} ((\boldsymbol{\lambda}_m^t)^\top \boldsymbol{R}_{m,p}^t) \boldsymbol{u}_p^t}_{=:\, \boldsymbol{B}}. \end{aligned} \tag{39}$$

Using the inequality $\|\boldsymbol{A} + \boldsymbol{B}\|^2 \leq 2\|\boldsymbol{A}\|^2 + 2\|\boldsymbol{B}\|^2$, we have:

$$\|\hat{\boldsymbol{g}}_{c,m}^t\|^2 \leq 2\|\boldsymbol{A}\|^2 + 2\|\boldsymbol{B}\|^2. \tag{40}$$

**Bounding Term $\boldsymbol{A}$:** Based on the fourth moment of Gaussian vectors, we have:

$$\mathbb{E}_{\mathcal{U}^t} \|\boldsymbol{A}\|^2 = \left( 1 + \frac{d_c + 1}{P} \right) \|\boldsymbol{g}_{c,m}^t\|^2. \tag{41}$$

**Bounding Term $B$:** From (36) and the definition of $\Gamma$, the remainder term is bounded by:

$$\|\boldsymbol{R}_{m,p}^t\| = \frac{\mu^2}{2}\|\mathcal{H}_{m,p}^t[\boldsymbol{u}_p^t, \boldsymbol{u}_p^t]\| \leq \frac{\mu^2}{2}\,\Gamma\,\|\boldsymbol{u}_p^t\|^2. \tag{42}$$

Consequently, the scalar projection is bounded by:

$$|(\boldsymbol{\lambda}_m^t)^\top \boldsymbol{R}_{m,p}^t| \leq \|\boldsymbol{\lambda}_m^t\|\|\boldsymbol{R}_{m,p}^t\| \leq \Gamma \cdot \frac{\mu^2}{2}\Gamma\|\boldsymbol{u}_p^t\|^2 = \frac{\mu^2}{2}\Gamma^2\|\boldsymbol{u}_p^t\|^2. \tag{43}$$

Using Jensen's inequality on the sum over $p$, we have:

$$\begin{aligned}
\mathbb{E}_{\mathcal{U}^t}\|\boldsymbol{B}\|^2 &\leq \mathbb{E}_{\mathcal{U}^t}\left[\frac{1}{P\mu^2}\sum_{p=1}^P \left((\boldsymbol{\lambda}_m^t)^\top \boldsymbol{R}_{m,p}^t\right)^2\|\boldsymbol{u}_p^t\|^2\right] \\
&\leq \frac{1}{\mu^2}\mathbb{E}_{\boldsymbol{u}}\left[\left(\frac{\mu^2}{2}\Gamma^2\|\boldsymbol{u}\|^2\right)^2\|\boldsymbol{u}\|^2\right] \\
&= \frac{\mu^2}{4}\Gamma^4\,\mathbb{E}_{\boldsymbol{u}}[\|\boldsymbol{u}\|^6],
\end{aligned} \tag{44}$$

where $\boldsymbol{u} \sim \mathcal{N}(\boldsymbol{0}, \boldsymbol{I}_{d_c})$ and the sixth moment of $\boldsymbol{u}$ is given by $\mathbb{E}[\|\boldsymbol{u}\|^6] = d_c(d_c + 2)(d_c + 4)$.

Combining (40), (41), and (44), we have:

$$\begin{aligned}
\mathbb{E}_{\mathcal{U}^t}\|\hat{\boldsymbol{g}}_{c,m}^t\|^2 &\leq 2\left(1 + \frac{d_c + 1}{P}\right)\|\boldsymbol{g}_{c,m}^t\|^2 + 2\left(\frac{\mu^2}{4}\Gamma^4 d_c(d_c + 2)(d_c + 4)\right) \\
&= C_1\|\boldsymbol{g}_{c,m}^t\|^2 + \frac{\mu^2}{2}d_c(d_c + 2)(d_c + 4)\Gamma^4.
\end{aligned} \tag{45}$$

This completes the proof. □

### A.4. Proof of Lemma A.3

**Lemma A.3** (Server-side Second Moment Bound). *Let $\boldsymbol{g}_s^t = \frac{1}{K}\sum_{m\in\mathcal{S}_t}\boldsymbol{g}_{s,m}^t$ be the aggregated server gradient. Under Assumption 4.2, the second moment of the server update is bounded by:*

$$\mathbb{E}_t\|\boldsymbol{g}_s^t\|^2 \leq \|\nabla_{\boldsymbol{\theta}_s}\mathcal{L}(\boldsymbol{\theta}^t)\|^2 + \Omega_s, \tag{46}$$

*where $\Omega_s := \frac{\sigma^2 + \kappa^2}{K}$ represents the server-side variance.*

*Proof.* We apply the bias–variance decomposition:

$$\begin{aligned}
\mathbb{E}_t\|\boldsymbol{g}_s^t\|^2 &= \|\mathbb{E}_t[\boldsymbol{g}_s^t]\|^2 + \mathbb{E}_t\|\boldsymbol{g}_s^t - \mathbb{E}_t[\boldsymbol{g}_s^t]\|^2 \\
&\overset{(a)}{=} \|\nabla_{\boldsymbol{\theta}_s}\mathcal{L}(\boldsymbol{\theta}^t)\|^2 + \mathbb{E}_t\|\boldsymbol{g}_s^t - \mathbb{E}_t[\boldsymbol{g}_s^t]\|^2.
\end{aligned}$$

where $(a)$ holds because $\mathbb{E}_t[\boldsymbol{g}_s^t] = \nabla_{\boldsymbol{\theta}_s}\mathcal{L}(\boldsymbol{\theta}^t)$ under Assumption 4.2.

We decompose the second term into two components: the intra-client data noise and the inter-client sampling noise:

$$\boldsymbol{g}_s^t - \nabla_{\boldsymbol{\theta}_s}\mathcal{L}(\boldsymbol{\theta}^t) = \underbrace{\frac{1}{K}\sum_{m\in\mathcal{S}_t}(\boldsymbol{g}_{s,m}^t - \nabla_{\boldsymbol{\theta}_s}\mathcal{L}_m(\boldsymbol{\theta}^t))}_{\text{Data Sampling Noise }\Delta_1^t} + \underbrace{\left(\frac{1}{K}\sum_{m\in\mathcal{S}_t}\nabla_{\boldsymbol{\theta}_s}\mathcal{L}_m(\boldsymbol{\theta}^t) - \nabla_{\boldsymbol{\theta}_s}\mathcal{L}(\boldsymbol{\theta}^t)\right)}_{\text{Client Sampling Noise }\Delta_2^t}. \tag{47}$$

Expanding the squared norm, the cross-term vanishes because the expectation of $\Delta_1^t$ is zero:

$$\mathbb{E}_t[\langle \Delta_1^t, \Delta_2^t\rangle] = \mathbb{E}_{\mathcal{S}_t}\left[\langle\mathbb{E}_{\boldsymbol{\xi}^t}[\Delta_1^t|\mathcal{S}_t], \Delta_2^t\rangle\right] = 0. \tag{48}$$

Thus, the variance decomposes into the sum of the individual variances:

$$\mathbb{E}_t \|g_s^t - \nabla_{\theta_s} \mathcal{L}(\theta^t)\|^2 = \mathbb{E}_t \|\Delta_1^t\|^2 + \mathbb{E}_{\mathcal{S}_t} \|\Delta_2^t\|^2. \tag{49}$$

By decoupling the expectations and expanding the squared norm, we obtain:

$$
\begin{aligned}
\mathbb{E}_t \|\Delta_1^t\|^2 &= \mathbb{E}_{\mathcal{S}_t} \left[ \mathbb{E}_{\xi^t} \left\| \frac{1}{K} \sum_{m \in \mathcal{S}_t} (g_{s,m}^t - \nabla_{\theta_s} \mathcal{L}_m(\theta^t)) \right\|^2 \right] \\
&\overset{(b)}{=} \mathbb{E}_{\mathcal{S}_t} \left[ \frac{1}{K^2} \sum_{m \in \mathcal{S}_t} \mathbb{E}_{\xi^t} \|g_{s,m}^t - \nabla_{\theta_s} \mathcal{L}_m(\theta^t)\|^2 \right] \\
&\overset{(c)}{\le} \mathbb{E}_{\mathcal{S}_t} \left[ \frac{1}{K^2} \sum_{m \in \mathcal{S}_t} \sigma^2 \right] \\
&= \frac{\sigma^2}{K},
\end{aligned}
$$

where $(b)$ holds because the data sampling is independent across clients and the local stochastic gradients are unbiased, causing the cross-terms to vanish; $(c)$ uses the Bounded Variance property in Assumption 4.2.

Next, we bound the client sampling noise term $\mathbb{E}_{\mathcal{S}_t} \|\Delta_2^t\|^2$ following a similar procedure:

$$
\begin{aligned}
\mathbb{E}_{\mathcal{S}_t} \|\Delta_2^t\|^2 &= \mathbb{E}_{\mathcal{S}_t} \left\| \frac{1}{K} \sum_{m \in \mathcal{S}_t} (\nabla_{\theta_s} \mathcal{L}_m(\theta^t) - \nabla_{\theta_s} \mathcal{L}(\theta^t)) \right\|^2 \\
&\overset{(d)}{=} \frac{1}{K^2} \sum_{m \in \mathcal{S}_t} \mathbb{E}_{\mathcal{S}_t} \|\nabla_{\theta_s} \mathcal{L}_m(\theta^t) - \nabla_{\theta_s} \mathcal{L}(\theta^t)\|^2 \\
&\overset{(e)}{\le} \frac{1}{K^2} \sum_{m \in \mathcal{S}_t} \kappa^2 \\
&= \frac{\kappa^2}{K},
\end{aligned}
$$

where $(d)$ is due to the independent and uniform sampling of clients, which ensures $\mathbb{E}_{\mathcal{S}_t}[\nabla_{\theta_s} \mathcal{L}_m(\theta^t)] = \nabla_{\theta_s} \mathcal{L}(\theta^t)$ and eliminates the cross-terms; $(e)$ applies the Bounded Gradient Dissimilarity property in Assumption 4.2.

Combining the squared bias and the variances derived above yields the final result:

$$\mathbb{E}_t \|g_s^t\|^2 \le \|\nabla_{\theta_s} \mathcal{L}(\theta^t)\|^2 + \frac{\sigma^2 + \kappa^2}{K}. \tag{50}$$

Defining $\Omega_s := \frac{\sigma^2 + \kappa^2}{K}$ completes the proof. $\qquad\square$

### A.5. Proof of Lemma A.4

**Lemma A.4** (Client-side Second Moment Bound). *Let $\hat{g}_c^t = \frac{1}{K} \sum_{m \in \mathcal{S}_t} \hat{g}_{c,m}^t$ be the aggregated client ZO gradient. Under Assumptions 4.1 and 4.2, the second moment of the client update is bounded by:*

$$\mathbb{E}_t \|\hat{g}_c^t\|^2 \le C_1 \|\nabla_{\theta_c} \mathcal{L}(\theta^t)\|^2 + \Omega_c, \tag{51}$$

*where $\Omega_c := C_1(\sigma^2 + \kappa^2) + \sigma_{ZO}^2$ collects the combined client-side variance, including stochastic gradient noise, client heterogeneity, and ZO estimation error.*

*Proof.* By Jensen's inequality and the law of iterated expectations, we obtain:

$$\mathbb{E}_t \left\| \hat{\boldsymbol{g}}_c^t \right\|^2 = \mathbb{E}_{\mathcal{S}_t} \left[ \mathbb{E}_{\boldsymbol{\xi}^t, \mathcal{U}^t} \left[ \left\| \frac{1}{K} \sum_{m \in \mathcal{S}_t} \hat{\boldsymbol{g}}_{c,m}^t \right\|^2 \Bigg| \mathcal{S}_t \right] \right]$$

$$\leq \mathbb{E}_{\mathcal{S}_t} \left[ \frac{1}{K} \sum_{m \in \mathcal{S}_t} \mathbb{E}_{\boldsymbol{\xi}^t, \mathcal{U}^t} \left[ \| \hat{\boldsymbol{g}}_{c,m}^t \|^2 \, \big| \, \mathcal{S}_t \right] \right]. \tag{52}$$

Applying Lemma A.2 and Assumption 4.2 yields:

$$\begin{aligned}
\mathbb{E}_{\boldsymbol{\xi}^t, \mathcal{U}^t} \left[ \| \hat{\boldsymbol{g}}_{c,m}^t \|^2 \, \big| \, \mathcal{S}_t \right] &= \mathbb{E}_{\xi_m^t} \left[ \mathbb{E}_{\mathcal{U}^t} \left[ \| \hat{\boldsymbol{g}}_{c,m}^t \|^2 \, \big| \, \xi_m^t, \mathcal{S}_t \right] \right] \\
&\leq \mathbb{E}_{\xi_m^t} \left[ C_1 \| \boldsymbol{g}_{c,m}^t \|^2 + \sigma_{ZO}^2 \right] \\
&\leq C_1 \left( \| \nabla_{\boldsymbol{\theta}_c} \mathcal{L}_m(\boldsymbol{\theta}^t) \|^2 + \sigma^2 \right) + \sigma_{ZO}^2.
\end{aligned} \tag{53}$$

Taking expectation over client sampling and using Bounded Gradient Dissimilarity from Assumption 4.2 yields:

$$\begin{aligned}
\mathbb{E}_t \| \hat{\boldsymbol{g}}_c^t \|^2 &\leq \frac{1}{M} \sum_{m=1}^{M} \left( C_1 \| \nabla_{\boldsymbol{\theta}_c} \mathcal{L}_m(\boldsymbol{\theta}^t) \|^2 + C_1 \sigma^2 + \sigma_{ZO}^2 \right) \\
&= C_1 \left( \frac{1}{M} \sum_{m=1}^{M} \| \nabla_{\boldsymbol{\theta}_c} \mathcal{L}_m(\boldsymbol{\theta}^t) \|^2 \right) + C_1 \sigma^2 + \sigma_{ZO}^2 \\
&= C_1 \left( \| \nabla_{\boldsymbol{\theta}_c} \mathcal{L}(\boldsymbol{\theta}^t) \|^2 + \frac{1}{M} \sum_{m=1}^{M} \| \nabla_{\boldsymbol{\theta}_c} \mathcal{L}_m(\boldsymbol{\theta}^t) - \nabla_{\boldsymbol{\theta}_c} \mathcal{L}(\boldsymbol{\theta}^t) \|^2 \right) + C_1 \sigma^2 + \sigma_{ZO}^2 \\
&\leq C_1 \left( \| \nabla_{\boldsymbol{\theta}_c} \mathcal{L}(\boldsymbol{\theta}^t) \|^2 + \kappa^2 \right) + C_1 \sigma^2 + \sigma_{ZO}^2 \\
&= C_1 \| \nabla_{\boldsymbol{\theta}_c} \mathcal{L}(\boldsymbol{\theta}^t) \|^2 + C_1 (\sigma^2 + \kappa^2) + \sigma_{ZO}^2.
\end{aligned} \tag{54}$$

Substituting the definition of $\Omega_c$, we obtain the stated bound. $\qquad \square$

## A.6. Proof of Lemma A.5

**Lemma A.5** (Local Bias of the ZO Gradient Estimator). *Condition on $(\mathcal{F}_t, \mathcal{S}_t, \xi_m^t)$, so that $\boldsymbol{\theta}_c^t$, $\boldsymbol{\lambda}_m^t$, and $\boldsymbol{g}_{c,m}^t$ are fixed. Let $\hat{\boldsymbol{g}}_{c,m}^t$ be the zero-order gradient estimate for client $m$. The bias of this estimator relative to the true local stochastic gradient $\boldsymbol{g}_{c,m}^t = \nabla_{\boldsymbol{\theta}_c} \ell(\boldsymbol{\theta}^t; \xi_m^t)$ satisfies:*

$$\| \mathbb{E}_{\mathcal{U}^t} [\hat{\boldsymbol{g}}_{c,m}^t] - \boldsymbol{g}_{c,m}^t \|^2 \leq \frac{\mu^2 \Gamma^4}{4} (d_c + 3)^3. \tag{55}$$

*Proof.* We aim to bound the norm of the bias vector $\boldsymbol{b}_m^t := \mathbb{E}_{\mathcal{U}^t} [\hat{\boldsymbol{g}}_{c,m}^t] - \boldsymbol{g}_{c,m}^t$.

Recall from the proof of Lemma A.2 (Eq. 39) that the estimator decomposes into a primary gradient term and a residual term:

$$\hat{\boldsymbol{g}}_{c,m}^t = \underbrace{\left( \frac{1}{P} \sum_{p=1}^{P} \boldsymbol{u}_p^t (\boldsymbol{u}_p^t)^\top \right) \boldsymbol{g}_{c,m}^t}_{\text{Gradient Term } \boldsymbol{A}} + \underbrace{\frac{1}{P\mu} \sum_{p=1}^{P} \delta_p^t \boldsymbol{u}_p^t}_{\text{Residual Term } \boldsymbol{B}}, \tag{56}$$

where the scalar residual is $\delta_p^t = (\boldsymbol{\lambda}_m^t)^\top \boldsymbol{R}_{p,m}^t$.

Taking the expectation with respect to the perturbations $\mathcal{U}^t = \{\boldsymbol{u}_p^t\}_{p=1}^{P}$:

1. For the Gradient Term $\boldsymbol{A}$, since $\mathbb{E}[\boldsymbol{u}_p^t (\boldsymbol{u}_p^t)^\top] = \boldsymbol{I}_{d_c}$, we have:

$$\mathbb{E}_{\mathcal{U}^t}[\boldsymbol{A}] = \boldsymbol{I}_{d_c} \boldsymbol{g}_{c,m}^t = \boldsymbol{g}_{c,m}^t. \tag{57}$$

2. For the Residual Term $\boldsymbol{B}$, due to the linearity of expectation and identical distribution of $\boldsymbol{u}_p^t$:

$$\mathbb{E}_{\mathcal{U}^t}[\boldsymbol{B}] = \frac{1}{\mu}\mathbb{E}_{\boldsymbol{u}^t}[\delta(\boldsymbol{u}^t)\boldsymbol{u}^t], \tag{58}$$

where $\boldsymbol{u}^t \sim \mathcal{N}(\boldsymbol{0}, \boldsymbol{I}_{d_c})$ and $\delta(\boldsymbol{u}^t) = (\boldsymbol{\lambda}_m^t)^\top \boldsymbol{R}(\boldsymbol{u}^t)$.

Thus, the local bias vector is strictly $\boldsymbol{b}_m^t = \frac{1}{\mu}\mathbb{E}_{\boldsymbol{u}^t}[\delta(\boldsymbol{u}^t)\boldsymbol{u}^t]$. Use Jensen's inequality:

$$\|\boldsymbol{b}_m^t\| \leq \frac{1}{\mu}\mathbb{E}_{\boldsymbol{u}^t}[|\delta(\boldsymbol{u}^t)|\|\boldsymbol{u}^t\|]. \tag{59}$$

From the proof in Lemma A.2, we have $\|\boldsymbol{\lambda}_m^t\| \leq \Gamma$ and $\|\boldsymbol{R}(\boldsymbol{u}^t)\| \leq \frac{\mu^2}{2}\Gamma\|\boldsymbol{u}^t\|^2$. Applying the Cauchy-Schwarz inequality:

$$|\delta(\boldsymbol{u}^t)| = |(\boldsymbol{\lambda}_m^t)^\top \boldsymbol{R}(\boldsymbol{u}^t)| \leq \|\boldsymbol{\lambda}_m^t\|\|\boldsymbol{R}(\boldsymbol{u}^t)\| \leq \Gamma \cdot \frac{\mu^2}{2}\Gamma\|\boldsymbol{u}^t\|^2 = \frac{\mu^2\Gamma^2}{2}\|\boldsymbol{u}^t\|^2. \tag{60}$$

Substituting this back into the norm bound for $\boldsymbol{b}_m^t$:

$$\|\boldsymbol{b}_m^t\| \leq \frac{1}{\mu}\mathbb{E}_{\boldsymbol{u}^t}\left[\left(\frac{\mu^2\Gamma^2}{2}\|\boldsymbol{u}^t\|^2\right)\|\boldsymbol{u}^t\|\right] \tag{61}$$

$$= \frac{\mu\Gamma^2}{2}\mathbb{E}_{\boldsymbol{u}^t}[\|\boldsymbol{u}^t\|^3]. \tag{62}$$

For a Gaussian vector $\boldsymbol{u}^t \sim \mathcal{N}(\boldsymbol{0}, \boldsymbol{I}_{d_c})$, the $k$-th moment of its norm satisfies $\mathbb{E}[\|\boldsymbol{u}^t\|^k] \leq (d_c + k)^{k/2}$. For $k = 3$:

$$\mathbb{E}_{\boldsymbol{u}^t}[\|\boldsymbol{u}^t\|^3] \leq (d_c + 3)^{3/2}. \tag{63}$$

Therefore:

$$\|\boldsymbol{b}_m^t\| \leq \frac{\mu\Gamma^2}{2}(d_c + 3)^{3/2}. \tag{64}$$

Finally, squaring both sides yields the bound on the squared norm of the bias:

$$\|\mathbb{E}_{\mathcal{U}^t}[\hat{\boldsymbol{g}}_{c,m}^t] - \boldsymbol{g}_{c,m}^t\|^2 = \|\boldsymbol{b}_m^t\|^2 \leq \frac{\mu^2\Gamma^4}{4}(d_c + 3)^3. \tag{65}$$

$\square$

### A.7. Proof of Lemma A.6

**Lemma A.6** (Global Bias Bound). *Under the same assumptions as Lemma A.5, the squared norm of the difference between the expected aggregated gradient estimator and the true global gradient is bounded by:*

$$\|\mathbb{E}_t[\hat{\boldsymbol{g}}_c^t] - \nabla_{\boldsymbol{\theta}_c}\mathcal{L}(\boldsymbol{\theta}^t)\|^2 \leq \frac{\mu^2\Gamma^4}{4}(d_c + 3)^3. \tag{66}$$

*Proof.* By the tower property,

$$\mathbb{E}_t[\hat{\boldsymbol{g}}_c^t] = \mathbb{E}_{\mathcal{S}_t}\left[\mathbb{E}_{\boldsymbol{\xi}^t, \mathcal{U}^t}\left[\frac{1}{K}\sum_{m \in \mathcal{S}_t}\hat{\boldsymbol{g}}_{c,m}^t \,\middle|\, \mathcal{S}_t\right]\right]$$

$$= \mathbb{E}_{\mathcal{S}_t}\left[\frac{1}{K}\sum_{m \in \mathcal{S}_t}\mathbb{E}_{\xi_m^t, \mathcal{U}^t}[\hat{\boldsymbol{g}}_{c,m}^t]\right], \tag{67}$$

where the second equality holds because $\hat{\boldsymbol{g}}_{c,m}^t$ depends only on $(\xi_m^t, \mathcal{U}^t)$ when conditioned on $(\mathcal{F}_t, \mathcal{S}_t)$.

For a specific client $m$, we define

$$\boldsymbol{b}_m^t := \mathbb{E}_{\mathcal{U}^t}\left[\hat{\boldsymbol{g}}_{c,m}^t \,\middle|\, \xi_m^t\right] - \boldsymbol{g}_{c,m}^t. \tag{68}$$

Then

$$
\begin{aligned}
\mathbb{E}_{\xi_m^t, \mathcal{U}^t}\left[\hat{\boldsymbol{g}}_{c,m}^t\right] &= \mathbb{E}_{\xi_m^t}\left[\mathbb{E}_{\mathcal{U}^t}\left[\hat{\boldsymbol{g}}_{c,m}^t \,\middle|\, \xi_m^t\right]\right] \\
&= \mathbb{E}_{\xi_m^t}\left[\boldsymbol{g}_{c,m}^t + \boldsymbol{b}_m^t\right] \\
&= \nabla_{\boldsymbol{\theta}_c}\mathcal{L}_m(\boldsymbol{\theta}^t) + \mathbb{E}_{\xi_m^t}[\boldsymbol{b}_m^t],
\end{aligned}
\tag{69}
$$

where the last line uses the unbiasedness part of Assumption 4.2. Substituting this identity back and using uniform client sampling, we obtain

$$\mathbb{E}_t[\hat{\boldsymbol{g}}_c^t] - \nabla_{\boldsymbol{\theta}_c}\mathcal{L}(\boldsymbol{\theta}^t) = \mathbb{E}_{\mathcal{S}_t}\left[\frac{1}{K}\sum_{m \in \mathcal{S}_t}\mathbb{E}_{\xi_m^t}[\boldsymbol{b}_m^t]\right]. \tag{70}$$

We now bound the squared norm of this difference:

$$
\begin{aligned}
\left\|\mathbb{E}_t[\hat{\boldsymbol{g}}_c^t] - \nabla_{\boldsymbol{\theta}_c}\mathcal{L}(\boldsymbol{\theta}^t)\right\|^2 &\le \mathbb{E}_{\mathcal{S}_t}\left[\left\|\frac{1}{K}\sum_{m \in \mathcal{S}_t}\mathbb{E}_{\xi_m^t}[\boldsymbol{b}_m^t]\right\|^2\right] \\
&\le \mathbb{E}_{\mathcal{S}_t}\left[\frac{1}{K}\sum_{m \in \mathcal{S}_t}\left\|\mathbb{E}_{\xi_m^t}[\boldsymbol{b}_m^t]\right\|^2\right] \\
&\le \mathbb{E}_{\mathcal{S}_t}\left[\frac{1}{K}\sum_{m \in \mathcal{S}_t}\mathbb{E}_{\xi_m^t}\left[\|\boldsymbol{b}_m^t\|^2\right]\right] \\
&\le \mathbb{E}_{\mathcal{S}_t}\left[\frac{1}{K}\sum_{m \in \mathcal{S}_t}\frac{\mu^2\Gamma^4}{4}(d_c + 3)^3\right] \\
&= \frac{\mu^2\Gamma^4}{4}(d_c + 3)^3.
\end{aligned}
\tag{71}
$$

The first three inequalities are Jensen's inequality, and the last one uses Lemma A.5. $\qquad\square$

## A.8. Proof of Theorem 4.4

**Theorem 4.4** (Convergence Bound of HO-SFL). *Suppose Assumptions 4.1 and 4.2 hold. Let the learning rate satisfy $\eta \le \frac{1}{2\beta C_1}$. For any $T \ge 1$, the average squared gradient norm of HO-SFL is bounded by:*

$$
\begin{aligned}
\frac{1}{T}\sum_{t=0}^{T-1}\mathbb{E}\left\|\nabla\mathcal{L}(\boldsymbol{\theta}^t)\right\|^2 &\le \frac{4\Delta_{\mathcal{L}}}{\eta T} + 2\eta\beta\left(\Omega_c + \Omega_s\right) \\
&\quad + \frac{\mu^2\Gamma^4}{2}(d_c + 3)^3,
\end{aligned}
\tag{26}
$$

*where $\Delta_{\mathcal{L}} = \mathcal{L}(\boldsymbol{\theta}^0) - \mathcal{L}^*$, $\Omega_s = \frac{\sigma^2 + \kappa^2}{K}$ represents the server-side variance, and $\Omega_c = C_1(\sigma^2 + \kappa^2) + \sigma_{ZO}^2$ denotes the aggregated client-side variance.*

*Proof.* By the $\beta$-smoothness of the objective function (Assumption 4.1), for the update $\boldsymbol{\theta}^{t+1} = \boldsymbol{\theta}^t - \eta[\hat{\boldsymbol{g}}_c^t; \boldsymbol{g}_s^t]$, we have:

$$\mathcal{L}(\boldsymbol{\theta}^{t+1}) \le \mathcal{L}(\boldsymbol{\theta}^t) - \eta\langle\nabla\mathcal{L}(\boldsymbol{\theta}^t), [\hat{\boldsymbol{g}}_c^t; \boldsymbol{g}_s^t]\rangle + \frac{\beta\eta^2}{2}\|[\hat{\boldsymbol{g}}_c^t; \boldsymbol{g}_s^t]\|^2. \tag{72}$$

Taking the conditional expectation $\mathbb{E}_t$ on both sides yields:

$$
\begin{aligned}
\mathbb{E}_t[\mathcal{L}(\boldsymbol{\theta}^{t+1})] &\le \mathcal{L}(\boldsymbol{\theta}^t) - \eta\langle\nabla_{\boldsymbol{\theta}_c}\mathcal{L}(\boldsymbol{\theta}^t), \mathbb{E}_t[\hat{\boldsymbol{g}}_c^t]\rangle - \eta\langle\nabla_{\boldsymbol{\theta}_s}\mathcal{L}(\boldsymbol{\theta}^t), \mathbb{E}_t[\boldsymbol{g}_s^t]\rangle \\
&\quad + \frac{\beta\eta^2}{2}\left(\mathbb{E}_t\|\hat{\boldsymbol{g}}_c^t\|^2 + \mathbb{E}_t\|\boldsymbol{g}_s^t\|^2\right).
\end{aligned}
\tag{73}
$$

For the server side, we have $\mathbb{E}_t[\boldsymbol{g}_s^t] = \nabla_{\boldsymbol{\theta}_s} \mathcal{L}(\boldsymbol{\theta}^t)$, which implies:

$$-\eta \langle \nabla_{\boldsymbol{\theta}_s} \mathcal{L}, \mathbb{E}_t[\boldsymbol{g}_s^t] \rangle = -\eta \|\nabla_{\boldsymbol{\theta}_s} \mathcal{L}(\boldsymbol{\theta}^t)\|^2. \tag{74}$$

For the client side, applying Young's inequality ($-\langle \boldsymbol{a}, \boldsymbol{b} \rangle \leq \frac{1}{2}\|\boldsymbol{a}\|^2 + \frac{1}{2}\|\boldsymbol{b}\|^2$) yields:

$$-\eta \langle \nabla_{\boldsymbol{\theta}_c} \mathcal{L}, \mathbb{E}_t[\hat{\boldsymbol{g}}_c^t] \rangle = -\eta \langle \nabla_{\boldsymbol{\theta}_c} \mathcal{L}, \nabla_{\boldsymbol{\theta}_c} \mathcal{L} + (\mathbb{E}_t[\hat{\boldsymbol{g}}_c^t] - \nabla_{\boldsymbol{\theta}_c} \mathcal{L}) \rangle \tag{75}$$

$$= -\eta \|\nabla_{\boldsymbol{\theta}_c} \mathcal{L}\|^2 - \eta \langle \nabla_{\boldsymbol{\theta}_c} \mathcal{L}, \mathbb{E}_t[\hat{\boldsymbol{g}}_c^t] - \nabla_{\boldsymbol{\theta}_c} \mathcal{L} \rangle \tag{76}$$

$$\leq -\frac{\eta}{2} \|\nabla_{\boldsymbol{\theta}_c} \mathcal{L}\|^2 + \frac{\eta}{2} \|\mathbb{E}_t[\hat{\boldsymbol{g}}_c^t] - \nabla_{\boldsymbol{\theta}_c} \mathcal{L}\|^2. \tag{77}$$

Combining these, the total linear term is bounded by:

$$-\eta \langle \nabla_{\boldsymbol{\theta}_c} \mathcal{L}, \mathbb{E}_t[\hat{\boldsymbol{g}}_c^t] \rangle - \eta \langle \nabla_{\boldsymbol{\theta}_s} \mathcal{L}, \mathbb{E}_t[\boldsymbol{g}_s^t] \rangle \leq -\eta \|\nabla_{\boldsymbol{\theta}_s} \mathcal{L}\|^2 - \frac{\eta}{2} \|\nabla_{\boldsymbol{\theta}_c} \mathcal{L}\|^2 + \frac{\eta}{2} \|\mathbb{E}_t[\hat{\boldsymbol{g}}_c^t] - \nabla_{\boldsymbol{\theta}_c} \mathcal{L}\|^2 \tag{78}$$

Next, we address the quadratic terms by substituting the second moment bounds from Lemmas A.3 and A.4. For the server gradients, we have $\mathbb{E}_t\|\boldsymbol{g}_s^t\|^2 \leq \|\nabla_{\boldsymbol{\theta}_s} \mathcal{L}(\boldsymbol{\theta}^t)\|^2 + \Omega_s$. For the client gradients, we have $\mathbb{E}_t\|\hat{\boldsymbol{g}}_c^t\|^2 \leq C_1\|\nabla_{\boldsymbol{\theta}_c} \mathcal{L}(\boldsymbol{\theta}^t)\|^2 + \Omega_c$. Substituting these bounds back into the descent inequality and applying the learning rate condition $\eta \leq \frac{1}{2\beta C_1}$, we have:

$$\mathbb{E}_t[\mathcal{L}(\boldsymbol{\theta}^{t+1})] \leq \mathcal{L}(\boldsymbol{\theta}^t) - \left(\eta - \frac{\beta\eta^2}{2}\right)\|\nabla_{\boldsymbol{\theta}_s} \mathcal{L}(\boldsymbol{\theta}^t)\|^2 - \left(\frac{\eta}{2} - \frac{\beta\eta^2 C_1}{2}\right)\|\nabla_{\boldsymbol{\theta}_c} \mathcal{L}(\boldsymbol{\theta}^t)\|^2$$

$$+ \frac{\eta}{2}\|\mathbb{E}_t[\hat{\boldsymbol{g}}_c^t] - \nabla_{\boldsymbol{\theta}_c} \mathcal{L}(\boldsymbol{\theta}^t)\|^2 + \frac{\beta\eta^2}{2}(\Omega_c + \Omega_s)$$

$$\overset{(a)}{\leq} \mathcal{L}(\boldsymbol{\theta}^t) - \frac{\eta}{2}\|\nabla_{\boldsymbol{\theta}_s} \mathcal{L}(\boldsymbol{\theta}^t)\|^2 - \frac{\eta}{4}\|\nabla_{\boldsymbol{\theta}_c} \mathcal{L}(\boldsymbol{\theta}^t)\|^2$$

$$+ \frac{\eta}{2}\|\mathbb{E}_t[\hat{\boldsymbol{g}}_c^t] - \nabla_{\boldsymbol{\theta}_c} \mathcal{L}(\boldsymbol{\theta}^t)\|^2 + \frac{\beta\eta^2}{2}(\Omega_c + \Omega_s)$$

$$\overset{(b)}{\leq} \mathcal{L}(\boldsymbol{\theta}^t) - \frac{\eta}{4}\|\nabla \mathcal{L}(\boldsymbol{\theta}^t)\|^2 + \frac{\eta}{2}\|\mathbb{E}_t[\hat{\boldsymbol{g}}_c^t] - \nabla_{\boldsymbol{\theta}_c} \mathcal{L}(\boldsymbol{\theta}^t)\|^2 + \frac{\beta\eta^2}{2}(\Omega_c + \Omega_s). \tag{79}$$

where $(a)$ holds because $\eta \leq \frac{1}{2\beta C_1}$ implies the server coefficient $\eta - \frac{\beta\eta^2}{2} \geq \frac{\eta}{2}$ and the client coefficient $\frac{\eta}{2} - \frac{\beta\eta^2 C_1}{2} \geq \frac{\eta}{4}$; and $(b)$ follows by relaxing $-\frac{\eta}{2}\|\nabla_{\boldsymbol{\theta}_s}\|^2$ to $-\frac{\eta}{4}\|\nabla_{\boldsymbol{\theta}_s}\|^2$ to combine with the client term, yielding the total gradient norm $-\frac{\eta}{4}\|\nabla \mathcal{L}\|^2$.

Simultaneously, we apply **Lemma A.6** to bound the squared norm of the bias term:

$$\|\mathbb{E}_t[\hat{\boldsymbol{g}}_c^t] - \nabla_{\boldsymbol{\theta}_c} \mathcal{L}(\boldsymbol{\theta}^t)\|^2 \leq \frac{\mu^2 \Gamma^4}{4}(d_c + 3)^3. \tag{80}$$

Plugging this bias bound and the variance terms $(\Omega_c + \Omega_s)$ into the descent inequality yields:

$$\mathbb{E}_t[\mathcal{L}(\boldsymbol{\theta}^{t+1})] \leq \mathcal{L}(\boldsymbol{\theta}^t) - \frac{\eta}{4}\|\nabla \mathcal{L}(\boldsymbol{\theta}^t)\|^2 + \frac{\eta}{2}\left[\frac{\mu^2 \Gamma^4}{4}(d_c + 3)^3\right] + \frac{\beta\eta^2}{2}(\Omega_c + \Omega_s). \tag{81}$$

Taking the total expectation $\mathbb{E}$ over the full history up to round $T - 1$, summing over $t$, and rearranging to isolate the gradient norm, we have:

$$\frac{\eta}{4}\sum_{t=0}^{T-1}\mathbb{E}\|\nabla \mathcal{L}(\boldsymbol{\theta}^t)\|^2 \leq \mathcal{L}(\boldsymbol{\theta}^0) - \mathbb{E}[\mathcal{L}(\boldsymbol{\theta}^T)] + \frac{T\eta}{2}\left[\frac{\mu^2 \Gamma^4}{4}(d_c + 3)^3\right] + \frac{T\beta\eta^2}{2}(\Omega_c + \Omega_s). \tag{82}$$

Finally, using the fact that $\mathcal{L}(\boldsymbol{\theta}^T) \geq \mathcal{L}^*$ and dividing both sides by $\frac{T\eta}{4}$, we obtain the final convergence rate:

$$\frac{1}{T}\sum_{t=0}^{T-1}\mathbb{E}\|\nabla \mathcal{L}(\boldsymbol{\theta}^t)\|^2 \leq \frac{4(\mathcal{L}(\boldsymbol{\theta}^0) - \mathcal{L}^*)}{\eta T} + 2\eta\beta(\Omega_c + \Omega_s) + \frac{\mu^2 \Gamma^4}{2}(d_c + 3)^3. \tag{83}$$

This completes the proof. $\qquad \square$

# B. Detailed Experimental Settings

## B.1. Dataset and Data Partitioning

**Vision Tasks.** We evaluate on CIFAR-10 and CIFAR-100, two standard image classification benchmarks with $32 \times 32$ color images. CIFAR-10 contains 10 classes with 50,000 training images and 10,000 test images, while CIFAR-100 contains 100 classes with 50,000 training images and 10,000 test images. For both datasets, we employ standard data augmentation including random cropping, horizontal flipping and mean/std normalization. For Non-IID settings, we partition the training data among clients using a Dirichlet distribution with concentration parameter $\alpha = 1$, which induces moderate label heterogeneity across clients.

**Language Tasks.** We evaluate on three natural language understanding tasks from the GLUE benchmark. SST-2 is a binary sentiment classification task (67,349 training examples / 872 validation examples / 1,821 test examples). RTE is a binary textual entailment task (2,490 training / 277 validation / 3,000 test). WSC is a coreference resolution task (554 training / 104 validation / 146 test). SQuAD is a reading comprehension task (87,599 training / 10,570 validation / 9,616 test). All language tasks are conducted in an IID setting.

## B.2. Model Architecture and Split Configuration

**CV Model.** We use a ResNet-18 model pre-trained on ImageNet. The model consists of 4 residual blocks. We split the model after the 2nd residual block. During training, all Batch Normalization (BN) layers are frozen to prevent statistics divergence in non-IID settings.

**LLM Models.** We consider standard causal language models (decoder-only Transformers) consisting of an embedding layer followed by a stack of Transformer blocks and a language modeling head. In the following, "layer" refers to the Transformer blocks. Unless otherwise specified, we used the following split configuration in our experiments:

- **OPT-125M:** Total 12 layers. Client holds the first 3 layers.
- **Gemma-3-270M:** Total 20 layers. Client holds the first 5 layers.
- **LLaMA-3.2-1B:** Total 18 layers. Client holds the first 5 layers.
- **LLaMA-3.2-3B:** Total 28 layers. Client holds the first 5 layers.
- **Qwen3-8B:** Total 36 layers. Client holds the first 5 layers.

## B.3. Baseline Introduction

- **SFL** (Thapa et al., 2022): A hybrid framework amalgamating FL and SL to enable parallel training across distributed clients. We implement the SFLv1 variant, which performs parallel server-side aggregation, and set the local epoch to 1.

- **SplitLoRA** (Lin et al., 2024): A first-order SFL framework for LLMs that combines standard split learning with LoRA, which utilizes full backpropagation on both client and server sides. In our experiments, we set the local epoch to 1.

- **FSL-SAGE** (Nair et al., 2025): A communication-efficient framework that employs client-side auxiliary models to estimate server gradients, allowing clients to perform multiple local updates before communicating with the server. We implement this baseline with a local epoch of 1 and a server update interval of 5, while setting the auxiliary model alignment interval to 1. For the ResNet-18 experiments, the auxiliary model is constructed by cascading the first server-side ResNet block with the final fully-connected layer.

- **ZO-SFL**: A memory-efficient baseline constructed by integrating zeroth-order optimization into the standard SFL architecture. Implementing the Simultaneous Perturbation Stochastic Approximation (SPSA) estimator (Malladi et al., 2023), ZO-SFL performs two forward passes per step with symmetric random perturbations $\pm\mu\mathbf{z}$ applied to the model parameters. The gradient is estimated via the central difference method: $\hat{\mathbf{g}} = \frac{\mathcal{L}(\boldsymbol{\theta}+\mu\mathbf{z})-\mathcal{L}(\boldsymbol{\theta}-\mu\mathbf{z})}{2\mu}\mathbf{z}$, enabling backpropagation-free training at the cost of dual activation transmissions. We set the perturbation magnitude $\mu = 10^{-3}$ and the local epoch to 1.

- **MU-SplitFed** (Liang et al., 2026): A straggler-resilient ZO framework that utilizes an unbalanced update mechanism to decouple client and server computations. We configure the server to perform $\tau = 2$ local updates for every client interaction to maximize server-side computational utilization.

### B.4. Hyperparameters

**Batch Size and Client Participation.** We use a batch size of 32 for both vision and language tasks. For vision experiments, we use 100 clients in total and sample 10 clients per round; for language experiments, we use 10 clients in total and sample 3 clients per round.

**Low-Rank Adaptation(LoRA).** We apply LoRA for language tasks with rank $r = 8$ and scaling factor $\alpha = 16$ to the query and value projection matrices.

**Optimizer.** For HO-SFL, SFL, SplitLoRA, FSL-SAGE, and ZO-SFL, we use AdamW with learning rate $1 \times 10^{-5}$, $\beta_1 = 0.9$, $\beta_2 = 0.999$, and weight decay $5 \times 10^{-4}$. For MU-SplitFed, we follow the original paper and use standard SGD without momentum, with client learning rate 0.005, server learning rate 0.01, and global learning rate 0.3.

**Perturbation & Smoothing.** Unless otherwise specified, we set $P = 5$ for vision tasks and $P = 2$ for language tasks, and use $\mu = 10^{-3}$.

### B.5. Fairness in Comparison.

In federated settings, *communication rounds* are often used as a proxy for the communication budget. However, in our comparison, this metric is not directly comparable across methods because a "round" corresponds to different numbers of client updates. Specifically, HO-SFL and MU-SplitFed perform client-side aggregation at every local step, whereas other methods aggregate after $E$ local epochs. Therefore, fixing the number of communication rounds would implicitly allow some methods to perform many more client updates than others, confounding optimization efficiency with update frequency.

To make the comparison fair, we use *proceeded samples* as the unified horizontal axis and stopping criterion. Under our implementation, a batch size of processed samples corresponds to one client update step (one forward-backward or ZO update, depending on the method). Thus, matching proceeded samples ensures that all methods perform the same number of client update steps and consume the same amount of training data, regardless of how often they communicate.

## C. Additional Results

### C.1. Latency Hiding Feasibility Analysis

To evaluate the feasibility of latency hiding in HO-SFL for resource-constrained devices, we simulated a split fine-tuning pipeline for the **LLaMA-3.2-1B** model. Our simulation adopts conservative parameters to reflect real-world 5G edge computing scenarios, where uplink bandwidth is often the primary bottleneck. The specific settings are detailed below:

- **Network Environment:** Based on global 5G experience reports (Narayanan et al., 2020), we model a standard 5G connection with an uplink bandwidth of 30 Mbps and a downlink bandwidth of 200 Mbps. The round-trip time (RTT) is set to 30 ms.

- **Client:** We simulate an edge AI accelerator with 2.0 TFLOPS (FP16) compute power. This is representative of current

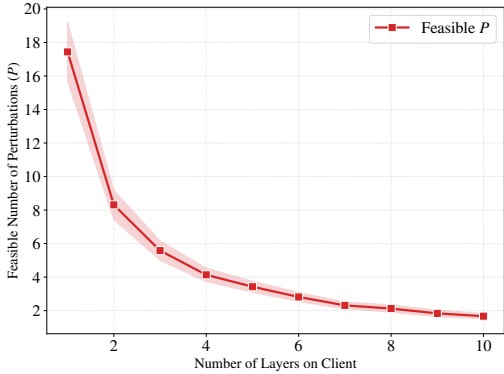

*Figure 5.* Feasibility analysis of latency hiding under constrained edge resources.

high-end embedded systems, such as the NVIDIA Jetson Orin Nano or flagship mobile NPUs.

- **Server:** The server is modeled as a cloud-grade NVIDIA A100 GPU with 312 TFLOPS (FP16) performance.

- **Workload:** We assume a batch size of 32 and a sequence length of 256. The simulation accounts for 10% random noise in network and computing speed to generate error bands.

Figure 5 illustrates the maximum number of perturbation passes ($P$) that can be fully overlapped within the communication and server-side processing latency. As the client-side model deepens, the local forward pass time consumes more of the fixed latency budget. However, for typical split learning configurations where the client retains the first few layers (e.g., 3-5 layers) to preserve data privacy, the system exhibits high tolerance. Specifically, with 4 client layers, the idle window accommodates approximately **4 perturbation passes** on average. This confirms that under typical 5G edge conditions, the communication bottleneck and server processing latency naturally mask the computational overhead of the multiple forward passes required by HO-SFL.

### C.2. More Results on Vision Tasks

In this section, we provide additional vision results to complement the main text. Figure 6 reports the validation accuracy curves on CIFAR-100 under both IID and Non-IID settings, while Table 3 summarizes the final test accuracy with standard deviation across CIFAR-10/100 and different data partitions.

*Table 3.* Performance comparison across different vision tasks. The notation $Acc_{(Std)}$ denotes the test accuracy (%) and its standard deviation. **Bold** indicates the best performance among comparative methods for each task.

| Task (Dataset) | First Order | | Zeroth Order | | Hybrid |
| --- | --- | --- | --- | --- | --- |
| | **SFL** | **FSL SAGE** | **ZO-SFL** | **Mu-SplitFed** | **HO-SFL (Ours)** |
| CIFAR-10 (IID) | $\mathbf{77.5}_{(\mathbf{0.5})}$ | $62.1_{(1.5)}$ | $12.3_{(1.0)}$ | $14.0_{(0.9)}$ | $75.0_{(0.3)}$ |
| CIFAR-10 (Non-IID) | $69.3_{(2.6)}$ | $37.9_{(2.9)}$ | $11.8_{(1.2)}$ | $13.6_{(1.0)}$ | $\mathbf{69.6}_{(\mathbf{0.8})}$ |
| CIFAR-100 (IID) | $43.2_{(0.4)}$ | $8.3_{(4.2)}$ | $1.2_{(0.2)}$ | $1.3_{(0.1)}$ | $\mathbf{44.2}_{(\mathbf{0.3})}$ |
| CIFAR-100 (Non-IID) | $39.5_{(0.6)}$ | $4.0_{(3.0)}$ | $1.2_{(0.1)}$ | $1.3_{(0.1)}$ | $\mathbf{42.5}_{(\mathbf{0.9})}$ |

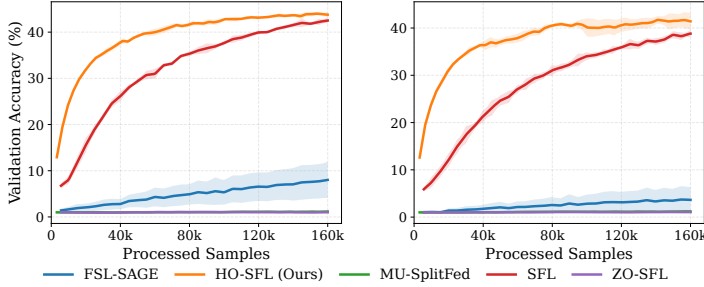

*Figure 6.* Validation accuracy convergence on CIFAR-100 under IID (left) and Non-IID (right) settings.

### C.3. Ablation Study on Client Model Depth

To investigate the impact of the client-side model complexity on HO-SFL's performance, we conducted an ablation study by varying the split point of the LLaMA-3.2-1B model on the SST-2 task. Specifically, we varied the number of transformer layers retained on the client side (denoted as "Cut Layer") from 2 to 8, thereby directly altering the dimensionality of the client-side parameters $d_c$.

As illustrated in Figure 7, while HO-SFL achieves convergence across all settings, we observe a consistent trend: as the number of client-side layers increases, the convergence speed slightly decelerates, and the final accuracy exhibits a marginal decline. For instance, the configuration with 2 client layers (Cut Layer 2) converges most rapidly to approx. 94.5%, whereas the 8-layer configuration (Cut Layer 8) saturates around 93.0%.

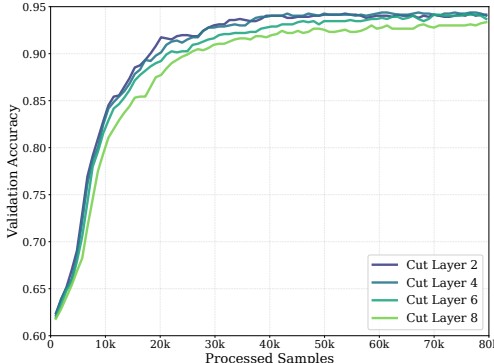

*Figure 7.* Impact of client-side model depth on convergence. We evaluate LLaMA-3.2-1B on the SST-2 task with varying numbers of transformer layers (2, 4, 6, 8) allocated to the client.

This empirical observation aligns perfectly with our theoretical analysis in Section 4. Recall that Corollary 4.5 establishes the convergence rate of HO-SFL as $\mathcal{O}(\sqrt{d_c/PT})$. Since the client employs zeroth-order optimization, the estimator's variance is inherently sensitive to the problem dimension $d_c$. Increasing the cut layer count significantly expands $d_c$, thereby introducing higher variance into the gradient estimation and slightly hindering optimization efficiency. Nevertheless, the performance degradation remains minimal even with 8 layers, confirming that our hybrid-order design effectively mitigates the severe dimension-dependent slowdown typically observed in pure ZO methods.

### C.4. Ablation Study on $P$ and $\mu$

To analyze the impact of the number of perturbations $P$ and the smoothing parameter $\mu$, we conduct ablation studies using the ResNet-18 on the CIFAR-10 dataset. The results are illustrated in Figure 8.

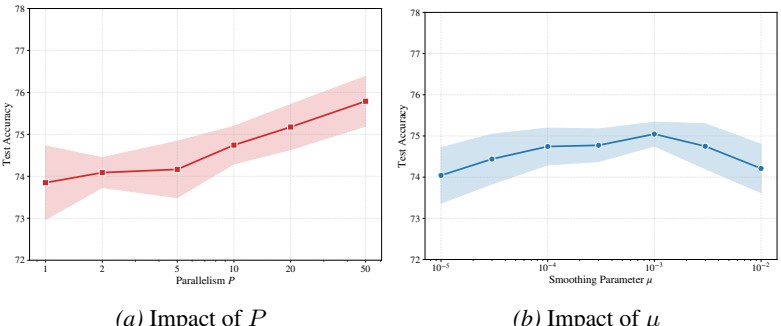

*(a)* Impact of $P$  *(b)* Impact of $\mu$

*Figure 8.* Ablation studies on CIFAR-10 with ResNet-18, focusing on the perturbation number $P$ and the smoothing parameter $\mu$. Solid lines denote the mean performance, and shaded regions represent the standard deviation across 10 random seeds.

**Impact of Perturbation Number $P$.** Figure 8a reports the test accuracy as $P$ increases from 1 to 50. We fix the smoothing parameter to $\mu = 10^{-4}$ in this ablation. Consistent with our theoretical analysis in Corollary 4.5, increasing $P$ yields a strictly monotonic improvement in model performance. Since HO-SFL only requires transmitting scalar projections, increasing $P$ incurs negligible communication overhead.

**Sensitivity to Smoothing Parameter $\mu$.** Figure 8b illustrates the impact of $\mu$, varying from $10^{-5}$ to $10^{-2}$. We fix the number of perturbations to $P = 10$ in this ablation. As $\mu$ increases beyond $10^{-3}$, performance degrades. This aligns with Proposition 4.3, which states that the bias of the zeroth-order estimator scales with $\mathcal{O}(\mu^2 d_c)$. Conversely, when $\mu$ becomes too small, we observe a drop in accuracy. While theory suggests that a smaller $\mu$ reduces bias, practical implementations are bounded by machine precision (Jongeneel et al., 2024), rendering the gradient estimate dominated by numerical noise rather than informative signals.

## C.5. Communication Breakdown Data

Table 4 provides the raw data used for the communication efficiency analysis, recorded in the CIFAR-10 IID experiment. The values represent the *total* communication cost (GB) over the entire training process.

*Table 4.* Breakdown of total communication cost (GB).

| Component | HO-SFL | SFL | FSL-SAGE | ZO-SFL | MU-SplitFed |
|---|---|---|---|---|---|
| Uplink (Act) | 1.22 | 1.26 | 0.29 | 2.52 | 3.67 |
| Uplink (Model) | 0.00 | 0.84 | 0.84 | 0.84 | 12.97 |
| Uplink (Scalar) | 0.0002 | 0.0 | 0.0 | 0.0 | 0.0 |
| Downlink (Grad) | 1.22 | 1.26 | 0.00 | 0.00 | 0.00 |
| Downlink (Model) | 0.00 | 0.84 | 2.58 | 0.84 | 12.97 |
| Downlink (Scalar/Seed) | 0.004 | 0.0 | 0.0 | 0.00002 | 0.00002 |

## C.6. Memory Profiling Methodology

To rigorously evaluate the baseline hardware requirements, we explicitly disable advanced memory-saving techniques, such as gradient accumulation and activation checkpointing (Chen et al., 2016). The batch size is fixed at 32 across all experiments. Crucially, to accurately isolate and quantify the specific memory footprint of the client-side model, we implement a specialized `Mocker` module. This module simulates the server's interaction by accepting uploaded activations and returning dummy gradients or loss scalars with matching dimensions, thereby decoupling the client's memory usage from server-side overhead. We utilize Nvidia's `nvidia-smi` command-line utility to monitor real-time GPU memory usage.

