# OpenReview forum: "HO-SFL: Hybrid-Order Split Federated Learning with Backprop-Free Clients and Dimension-Free Aggregation"
_ICML.cc/2026/Conference — ICML 2026 regular_

### Official Review · Reviewer_xP6T · 2026-03-01

**Soundness:** 3
**Presentation:** 3
**Significance:** 3
**Originality:** 3
**Overall Recommendation:** 4
**Confidence:** 4

**Summary:**

This paper proposes Hybrid-Order Split Federated Learning (HO-SFL) to address the severe memory constraints of fine-tuning large models on edge devices, a challenge where standard Split Federated Learning (SFL) still requires memory-intensive backpropagation (BP) on clients, and purely zeroth-order (ZO) methods suffer from prohibitively slow, dimension-dependent convergence. To resolve this dilemma, HO-SFL reformulates the SFL process within a Lagrangian framework to decouple the optimization landscape. In this hybrid setup, the resource-rich server performs first-order updates via BP, while the edge clients utilize server-transmitted activation gradients to conduct memory-efficient ZO optimization. Furthermore, by leveraging shared random seeds, HO-SFL allows clients to upload only a few scalar projections instead of full high-dimensional model weights, achieving communication-efficient dimension-free aggregation. Theoretically, the authors prove that HO-SFL achieves a convergence rate of $\mathcal{O}(\sqrt{d_{c}/PT})$, demonstrating that the framework successfully isolates the ZO convergence penalty to the significantly smaller client-side dimension $d_{c}$, thereby mitigating the curse of dimensionality typically seen in ZO methods. Empirically, extensive experiments across vision (ResNet-18) and language (OPT, Gemma, LLaMA) tasks validate that HO-SFL matches the convergence speed of first-order baselines while reducing client-side peak memory consumption to inference-only levels and drastically lowering communication costs.

**Compliance With Llm Reviewing Policy:**

Affirmed.

**Final Justification:**

I have no further questions and will keep my positive score.

**Key Questions For Authors:**

1. Could the authors evaluate HO-SFL against SFL baselines integrated with standard FL drift-correction methods (e.g., SCAFFOLD or FedNova) in Non-IID settings?


2. Could the authors provide a convergence comparison between HO-SFL and a standard full-model ZO-FL baseline (e.g., FedZO or DeComFL) on at least one language task?


3. Can the authors provide ablation studies for $P$ and $\mu$ on an LLM task (e.g., LLaMA-3.2-1B), particularly observing the impact of numerical noise under FP16/BF16 precision?

**Limitations:**

yes

**Strengths And Weaknesses:**

Strengths:

1. The paper tackles the memory bottleneck in Split Federated Learning (SFL) by cleverly reformulating the optimization problem using a Lagrangian framework. This cleanly decouples the global optimization, allowing the resource-rich server to perform exact first-order updates (BP) while edge clients execute memory-efficient zeroth-order (ZO) updates.


2.  The authors provide a rigorous non-convex convergence analysis, proving that HO-SFL achieves a convergence rate of $\mathcal{O}(\sqrt{d_c/PT})$. This theoretically justifies how the framework effectively isolates the ZO convergence penalty to the significantly smaller client-side dimension $d_c$, successfully mitigating the "curse of dimensionality" typically inherent in standard ZO optimization.


3. The empirical evaluation is solid, demonstrating effectiveness across both vision models (ResNet-18) and modern Large Language Models (OPT-125M, Gemma-3-270M, LLaMA-3.2-1B), proving that the method matches or closely approximates first-order baselines.


Weaknesses:

1. While the paper properly evaluates Non-IID settings for vision tasks , the authors state in Appendix B.1 that "All language tasks are conducted in an IID setting.". Given that data heterogeneity (Non-IID) is a fundamental challenge in federated learning that frequently causes severe client drift, the absence of Non-IID experiments for LLMs limits the real-world impact claims of the paper.


2. The ablation study on the number of perturbations $P$ is exclusively conducted on the small-scale vision task (ResNet-18 on CIFAR-10). For language tasks, the authors fix $P=2$ without providing an ablation. Given that LLM forward passes are significantly more computationally expensive, understanding the performance-efficiency trade-off of $P$ on LLMs is crucial. If LLMs require a larger $P$ to bridge the accuracy gap, the proposed "latency hiding" mechanism  might be violated, breaking the strict compute budget of edge devices.


3. The authors highlight that HO-SFL demonstrates superior robustness in Non-IID settings compared to standard SFL, attributing this to its step-wise aggregation avoiding client drift. However, comparing against vanilla SFL in Non-IID scenarios is a weak baseline. The federated learning community has long established methods specifically designed to tackle client drift, such as SCAFFOLD [1] or FedNova [2].

[1] Sai Praneeth Karimireddy, Satyen Kale, Mehryar Mohri, Sashank J. Reddi, Sebastian U. Stich, and Ananda Theertha Suresh. 2020. SCAFFOLD: stochastic controlled averaging for federated learning. In Proceedings of the 37th International Conference on Machine Learning (ICML'20), Vol. 119. JMLR.org, Article 476, 5132–5143.

[2] Jianyu Wang, Qinghua Liu, Hao Liang, Gauri Joshi, and H. Vincent Poor. 2020. Tackling the objective inconsistency problem in heterogeneous federated optimization. In Proceedings of the 34th International Conference on Neural Information Processing Systems (NIPS '20). Curran Associates Inc., Red Hook, NY, USA, Article 638, 7611–7623.

---

> ### Author Rebuttal · Authors · 2026-03-31
>
> We thank the reviewer for the thorough review and constructive suggestions on experiments. We address your questions below.
>
> > W1: Non-IID Settings for LLMs
>
> We agree that demonstrating Non-IID robustness for LLMs is critical. To address this concern, we have conducted the Non-IID experiments for the LLaMA-3.2-1B model. Following the setup in vision tasks, we applied Dirichlet partitioning ($\alpha=1$):
>
> | **Dataset** | **HO-SFL (IID)** | **HO-SFL (Non-IID)** |
> | ----------- | ---------------- | -------------------- |
> | **SST-2**   | 93.9%            | 94.0%                |
> | **WSC**     | 61.5%            | 63.5%                |
> | **RTE**     | 73.3%            | 72.9%                |
>
> HO-SFL maintains stable performance under Non-IID conditions. We believe the robust performance comes from the **dimension-free scalar aggregation at every single step**. This low-cost synchronization allows HO-SFL to mimic centralized training dynamics and thus mitigates client drift.
>
> > Q1 & W3: Comparison with Drift-Correction Methods
>
> We integrated FedProx into the standard SFL baseline. The table below summarizes the results across 10 random seeds:
>
> | **Task (Dataset)**  | **SFL**   | **SFL + FedProx** | **HO-SFL (Ours)** |
> | ------------------- | --------- | ----------------- | ----------------- |
> | CIFAR-10 (IID)      | **77.5%** | 77.3%             | 75.0%             |
> | CIFAR-10 (Non-IID)  | 69.3%     | **69.8%**         | 69.6%             |
> | CIFAR-100 (IID)     | 43.2%     | 42.9%             | **44.2%**         |
> | CIFAR-100 (Non-IID) | 39.5%     | 40.7%             | **42.5%**         |
>
> FedProx provides only marginal gains over SFL under NON-IID conditions. As discussed in W1, the step-wise aggregation of HO-SFL effectively prevents client drift.
>
> > Q2:  Comparison with Full-Model ZO-FL Baseline
>
> We compared HO-SFL with **DeComFL** (a SOTA ZO-FL baseline) by fine-tuning the LLaMA-3.2-1B model on the SST-2 task (IID partitioning), where clients hold 5 transformer blocks. The results are summarized below:
>
> | **Method**     | **Accuracy (%)** | **Peak Memory (Seq Len = 512)** |
> | -------------- | ---------------- | ------------------------------- |
> | **SplitLoRA** | 94.4%            | 17.09 GB                        |
> | **DeComFL**    | 81.9%            | 15.41 GB                        |
> | **HO-SFL**     | 93.9%            | 5.58 GB                         |
>
> 1. **Convergence:** DeComFL underperforms because of high gradient bias and variance introduced by the pure ZO estimator. By isolating the ZO to the client-side model, HO-SFL achieves results competitive with the pure FO methods.
> 2. **Memory:** DeComFL’s memory footprint reaches 15.41 GB, and HO-SFL keeps client memory at a lightweight level (5.58 GB).
>
> > Q3 & W2 : Ablation on $P$ and $\mu$ under BF16 Precision on LLMs
>
> As requested, we conducted experiments on the OPT-125M for the SST-2 task:
>
> Table A: Impact of Smoothing Parameter $\mu$  (Fixed  $P$ = 2)
>
> | **Precision** | $\mu$ = 1e-2 | $\mu$ = 1e-3 | $\mu$ = 1e-4 | $\mu$ = 1e-5 |
> | ------------- | ------------ | ------------ | ------------ | ------------ |
> | **FP32**      | 87.63%       | 87.65%       | 87.64%       | 87.61%       |
> | **BF16**      | 86.65%       | 86.66%       | 86.62%       | 86.68%       |
>
> Table B: Impact of Perturbation Number  $P$  (Fixed $\mu$ = 1e-3)
>
> | **Precision** | **$P$ = 1** | **$P$ = 2** | **$P$ = 5** | **$P$ = 10** |
> | ------------- | --------- | --------- | --------- | ---------- |
> | **FP32**      | 87.61%    | 87.65%    | 87.68%    | 87.73%     |
> | **BF16**      | 86.48%    | 86.57%    | 86.66%    | 86.66%     |
>
> *(Note: Values represent the mean accuracy across 10 random seeds.)*
>
> We highlight two observations:
>
> - We observe a minor performance drop of approximately 1% when shifting to BF16 precision.
> - HO-SFL demonstrates remarkable stability across a wide range of $\mu$ (from 1e-2 to 1e-5) and $P$ (from 1 to 10) under BF16.
>
> Thank you again for your time and effort in reviewing our paper. We hope our explanations clarify the points raised. We will reflect all the above discussions and additional results in the revised version of our manuscript.

---

> > ### Author Rebuttal · Reviewer_xP6T · 2026-04-02
> >
> > I have no further questions and will keep my positive score.

---

> > > ### Author Response · Authors · 2026-04-04
> > >
> > > Dear Reviewer xP6T,
> > >
> > > Thank you for your positive feedback and for confirming that we have fully addressed your questions.
> > >
> > > Your constructive feedback guided us to conduct new experiments, which make the paper's contribution more solid. Given that these additions address the initial limitations you identified, we would be incredibly grateful if you consider whether these additions strengthen the paper enough to warrant a higher score in your final evaluation.
> > >
> > > If any concerns remain, we are more than happy to discuss them further.
> > >
> > > Best regards,
> > >
> > > Authors

---

### Official Review · Reviewer_9fYv · 2026-03-11

**Soundness:** 3
**Presentation:** 3
**Significance:** 3
**Originality:** 2
**Overall Recommendation:** 4
**Confidence:** 3

**Summary:**

This paper proposes Hybrid-Order Split Federated Learning (HO-SFL), a distributed learning framework designed to address the memory and communication bottlenecks of training large models on resource-constrained edge devices. Traditional FL requires clients to perform full backpropagation (BP) locally, which becomes infeasible for modern large models due to the memory cost of storing intermediate activations. Split Federated Learning (SFL) partially mitigates this by splitting the model between clients and the server, but still requires clients to perform BP on their local model segments. HO-SFL introduces a hybrid-order optimization scheme: server side performs first-order optimization using standard backpropagation; client side performs zeroth-order (ZO) optimization, eliminating the need for gradient computation and reducing memory usage.

**Compliance With Llm Reviewing Policy:**

Affirmed.

**Final Justification:**

Thanks for the rebuttal. I keep my positive score.

**Key Questions For Authors:**

1. Zeroth-order methods typically require multiple function evaluations to estimate gradients. How does the overall training time of HO-SFL compare with standard SFL?

2. We know that ZO is sensitive to noise or pertubation number. How sensitive is HO-SFL to the number of perturbation samples? Is there a principled way to select this parameter? Or it is less sensitive because the split setting with only a fewer layers in client's side.

**Limitations:**

The authors should include discussion for the limitations or at least tradeoffs, such as training time.

**Strengths And Weaknesses:**

Strengths:
1. The paper provides formal convergence guarantees under standard assumptions (e.g., smoothness, bounded variance).

Weaknesses:
1. While the combination of ideas is interesting, each individual component has prior work: zeroth-order optimization for distributed learning, split federated learning, and dimension-reduced communication. The novelty mainly lies in integrating these components through a hybrid-order formulation, which may be perceived as less novel.

2. ZO optimization typically requires multiple function evaluations per update, which may significantly increase computation time. The paper should more clearly analyze: the trade-off between memory savings and runtime cost.

---

> ### Author Rebuttal · Authors · 2026-03-31
>
> Thank you for the constructive feedback. We are glad you recognized our convergence guarantees. We address your questions below.
>
> > W1: Novelty of HO-SFL
>
> We agree that the ingredients of HO-SFL exist in prior works. However, their integration is far from a trivial combination. The main challenge is that client-side and server-side parameters are inherently coupled in SFL. Directly introducing ZO to SFL leads to poor convergence and $P$-fold increase in uplink traffic (as seen in SFL-ZO). In contrast, our Lagrangian reformulation decouples the problem into two subproblems, allowing client-side ZO and server-side BP to naturally coexist.
>
> > W2, Q1 & Limitations: Training Time vs. Standard SFL
>
> We agree that ZO methods typically increase computation time due to multiple function evaluations. However, HO-SFL mitigates this through **latency hiding and dimension-free aggregation** discussed in Section 3.4.
>
> To demonstrate this better, we added a runtime simulation(https://anonymous.4open.science/r/HO-SFL-Figures/runtime_comparison.png) under the same edge settings as Appendix C.1. Our simulation suggests that much of the extra ZO forward cost can be hidden in realistic edge settings. This efficiency comes from two designs:
>
> 1. **Latency Hiding:** SFL’s idle window can fully cover (or partially cover, if $P$ exceeds the threshold) the execution time of HO-SFL’s $P$ local forward passes.
> 2. **Dimension-Free Aggregation:** Unlike SFL, which must transmit high-dimensional parameters, HO-SFL only transmits scalars. This largely decreases the communication overhead, further decreasing the training time.
>
> > Q2: Sensitivity to Perturbation Number and Selection Strategy
>
> HO-SFL is less sensitive to $P$ than pure ZO methods because our estimator’s variance scales only with the client-side dimension $d_c$ (Corollary 4.5), rather than the full model dimension.
>
> - **Sensitivity:** As shown in Appendix C.4 (Figure 8a) and our additional ablations (detailed in response to Reviewer xP6T’s Q3), while increasing P monotonically improves accuracy, the gain is **marginal rather than dramatic**.
> - **Principled Selection:** In practice, we recommend selecting $P$ based on the client’s hardware capacities to exploit the “free lunch” effect. As illustrated in the runtime figure (https://anonymous.4open.science/r/HO-SFL-Figures/runtime_comparison.png), the principled choice is to set P to the maximum value before the runtime curve bends upwards (the latency-hiding bound). By setting $P$ to this threshold, we improve the gradient estimation and convergence stability without increasing the end-to-end training time.
>
> Thank you for your time; your feedback helps us clarify these practical deployment details. We will highlight this selection principle in the revised version.

---

> > ### Author Rebuttal · Reviewer_9fYv · 2026-04-01
> >
> > I will keep my positive score.

---

> > > ### Author Response · Authors · 2026-04-04
> > >
> > > Dear Reviewer 9fYv,
> > >
> > > Thank you for your response and for the time you dedicated to reviewing our paper. We are glad to hear that your concerns have been fully resolved. Given this, we would be grateful if you would consider this resolution could be reflected in your final evaluation of the paper. If any questions or remaining concerns persist, we would be very happy to provide further clarification, as we value the review process as an invaluable opportunity to improve the clarity of the paper.
> > >
> > > Best regards,
> > >
> > > Authors

---

### Official Review · Reviewer_dssz · 2026-03-12

**Soundness:** 2
**Presentation:** 2
**Significance:** 1
**Originality:** 1
**Overall Recommendation:** 2
**Confidence:** 3

**Summary:**

This paper proposes HO-SFL for split federated learning (SFL), combining server-side first-order (FO) updates with client-side zeroth-order (ZO) updates via a Lagrangian formulation, and using activation-gradient feedback and scalar aggregation to enable BP-free client training with low communication overhead.

**Compliance With Llm Reviewing Policy:**

Affirmed.

**Final Justification:**

I appreciate the additional explanations and clarifications, and I agree that they are useful in better positioning the paper. The response partially addresses several of my concerns by

- explaining the role of the Lagrangian reformulation, and
- discussing practical tradeoffs against some engineering alternatives.

However, in my view, the rebuttal does not fully resolve the core issues I raised.

In particular, my main concern about novelty remains only partially addressed. I understand the authors' argument that combining server-side first-order updates with client-side zeroth-order updates in split federated learning is non-trivial, and that the Lagrangian formulation helps decouple the problem in a principled way. Still, I remain unconvinced that this amounts to a substantially new optimization paradigm.

I also find the additional discussion of experiments and language-task analysis helpful, but it does not fundamentally address my concern that the empirical evidence remains somewhat narrow relative to the breadth of the paper’s claims. Similarly, clarifying module-level roles improves readability but does not fully replace direct module-level isolation. On the theory side, the rebuttal helps explain what is formalized and why it is relevant, yet I am still not fully convinced that the theoretical contribution is strong enough beyond the specific hybrid setting considered.

Moreover, I think the experiment part is inadequate because it is limited to classification tasks rather than next-token decoding tasks, as classic ZO papers do, which could affect practicability.

In addition, I'm skeptical about the ZO-SFL baseline performance. The ZO-SFT OPT-1.3B accuracy on the GLUE subset is surprisingly low.
- Reported in submission: SST-2 / WSC / RTE = 52.8 / 36.5 / 52.0
- Reported in DeComFL: SST-2 / WSC / RTE = 85.1 / 59.6 / 57.1

I am aware that DeComFL is a general federated learning setting, and that the paper discussed split learning, but I still find the gap too wide. I noticed that authors use the same learning rate across all optimizers, which, to me, is not a good idea. Normally, ZO baselines require a learning rate grid search, as in Malladi et al., Zhao et al., and Li et al. (Li et al. did not report a grid search, but they report a different learning rate for each dataset, which implies careful hyperparameter selection). The current ZO baseline is somewhat undermined to me.

Overall, I think the current paper and rebuttal together make the work clearer and somewhat stronger, and I acknowledge that some concerns have been mitigated. However, the central concerns—especially those regarding novelty, experimental scope, and the scalability/generality of the proposed framework—are not fully resolved to my satisfaction. Therefore, before any further discussion, I would prefer to maintain my original score.

---
**References**

Malladi S, Gao T, Nichani E, et al. Fine-tuning language models with just forward passes[J]. Advances in Neural Information Processing Systems, 2023, 36: 53038-53075.

Li, Zhe, et al. “Achieving dimension-free communication in federated learning via zeroth-order optimization.” ICLR 2025.

Zhao Y, Dang S, Ye H, et al. "Second-order fine-tuning without pain for llms: A hessian informed zeroth-order optimizer." ICLR 2025

**Key Questions For Authors:**

[Q1] Please add comparisons against more direct engineering alternatives (e.g., smaller client splits and activation checkpointing/recomputation) to clarify the practical regime where HO-SFL is actually preferable.

[Q2] Appendix C.3 and C.4 should be described as parameter sensitivity analyses rather than ablation studies.

**Limitations:**

No.

The work did not compare the **trained performance** and **communication overhead** with pure ZO-FedSGD methods like [r1, r2]. The key limitation of the proposed method is that it requires uploading or downloading dense activations, resulting in **GBs** of data traffic (Figure 4). However, pure ZO-FedSGD requires only **KBs** of data traffic and no BP on the server. The reported performance does not justify its substantial additional communication and computation complexity.

[r1] Qin et al. Federated Full-Parameter Tuning of Billion-Sized Language Models with Communication Cost under 18 Kilobytes.

[r2] Xu et al. FwdLLM: Efficient FedLLM using Forward Gradient.

**Strengths And Weaknesses:**

**Strengths**

The motivation is reasonable and has practical value.

**Weaknesses**

[W1] Limited novelty: The core FO-on-server / ZO-on-client decomposition is an obvious engineering choice under SFL constraints, and the paper’s λ is not a new signal but simply the standard split-learning activation gradient reinterpreted in Lagrangian form.

[W2] Insufficient experimental strength: The main experiments focus primarily on CIFAR-10/100 and a small set of GLUE-style NLU tasks (SST-2/RTE/WSC), which is insufficient to support strong claims about edge-side large-model fine-tuning.

[W3] Insufficient language-task analysis: Although the paper reports language-task results, key analyses (especially split-depth effects and $P$,$μ$ parameter sensitivity) are not systematically shown across all language tasks, and the main paper provides only brief summary statements.

[W4] Unclear module-level contribution: The comparisons include FO/ZO baselines, but there is no direct module-level ablation to isolate which components (hybrid-order design, scalar aggregation, synchronization) drive the gains.

[W5] Limited theoretical depth: The theory mainly provides a convergence analysis/formalization of the hybrid design, without introducing a new theoretical problem or substantially stronger guarantees, and the rate still depends on the client-side dimension $d_c$.

[W6] Unclear practical scope: The paper does not clearly establish when this method is preferable to more direct alternatives (e.g., smaller client splits, activation checkpointing/recomputation, or other PEFT/splitting strategies).

---

> ### Author Rebuttal · Authors · 2026-03-31
>
> We thank the reviewer for their rigorous assessment and detailed evaluation of our paper.
>
> > W1: Novelty of HO-SFL and the Role of $\lambda$
>
> We agree that the motivation for FO-on-server / ZO-on-client is intuitive. However, **realizing** this idea is non-trivial. The main challenge is that client-side and server-side parameters are coupled in SFL. Our Lagrangian reformulation decouples the problem into two subproblems. This allows us to isolate the ZO bias to the smaller client-side subspace, bridging the gap between an intuitive idea and a viable algorithm.
>
> Furthermore, $\lambda$ is mathematically derived as the activation gradient. And we argue this equivalence is actually **an advantage**: it guarantees that HO-SFL’s server-side operations are identical to standard SFL, which prevents any additional communication and computing costs on the server and allows HO-SFL to integrate server-side innovations like GAS [1].
>
> > W2 & W3: Experimental Scope and Language Task Analysis
>
> Our experimental setup follows related literature (e.g., [2] and [3]) and we argue that sub-billion LLM fine-tuning task reflects edge deployment challenges [4].
>
> Regarding split-depth effects, we have demonstrated them in Appendix C.3. For the parameter sensitivity of $P$ and $\mu$ in the language task, please refer to our response to Reviewer xP6T’s Q3.
>
> > W4: Clarification of Module-Level Contributions
>
> We respectfully clarify that these components are not independent plug-and-play modules and their contributions can be isolated by the metrics they impact.
>
> - **Hybrid-Order Design** is the **only** component that influences training dynamics and memory footprint. Its contribution is isolated by accuracy and memory profiling.
> - **Scalar Aggregation** does not alter training dynamics or memory. Its contribution is isolated by our communication profiling. Moreover, it depends on the ZO scalar projections and cannot be implemented in standard SFL.
> - **Synchronization** is a system-level requirement under partial participation and is implemented across all baselines. In HO-SFL, it is naturally integrated with scalar aggregation.
>
> > W5: Theoretical Contribution
>
> We highlight that our theory is not a routine convergence analysis for a standard setting: it studies a *layer-wise* hybrid FO-ZO optimization landscape induced by the Lagrangian decoupling. To the best of our knowledge, this is a new problem and most related works analyze pure ZO.
>
> Regarding the dependence on the client-side dimension ($d_c$), we kindly note that the variance of ZO estimators intrinsically depends on the dimension of the perturbation space. Although ZO estimators cannot entirely eliminate dimension dependence, HO-SFL achieves FO-level convergence by isolating it strictly to the smaller client-side subspace.
>
> > Q1 & W6: Engineering Alternatives
>
> - **Shallow Splits:** Our memory analysis (https://anonymous.4open.science/r/HO-SFL-Figures/memory_comparison.png) shows that even with shallow client splits, standard SFL’s memory inflates rapidly as sequence length scales. In contrast, HO-SFL strictly maintains inference-level memory footprint, enabling training with a longer context window on constrained clients.
> - **Activation Checkpointing:** While this technique reduces activation storage, the peak memory footprint still exceeds inference-level and recomputing introduces substantial training latency. In HO-SFL, our ZO forward passes achieve an inference-level memory and can be masked via latency hiding (Section 3.4).
>
> These alternatives require **memory-intensive BP** and do not provide **dimension-free aggregation**. Some alternatives like shallow splits or PEFT can be integrated into HO-SFL to provide cumulative benefits, but cannot replace the fundamental memory and communication efficiency gained by HO-SFL.
>
> > Q2: Terminology Correction
>
> Thank you for the correction. We will change relevant content to “Parameter Sensitivity Analysis” in our revision.
>
> > Limitations: Comparison with Pure ZO-FL
>
> Due to space limits, please refer to our response to Reviewer xP6T (Q2) for comparison with DeComFL.
>
> While pure ZO-FL minimizes communication, it leads to high memory overhead and degraded convergence. For modern edge scenarios, where communication resources are relatively abundant but memory is the bottleneck, HO-SFL provides a more practical solution.
>
> We appreciate the time the reviewer took to evaluate our paper. We hope our rebuttal address your concerns.
>
> ---
>
> >  References
>
> [1] Yang, Jiarong, and Yuan Liu. “Gas: Generative activation-aided asynchronous split federated learning.” AAAI 2025.
>
> [2] Li, Zhe, et al. “Achieving dimension-free communication in federated learning via zeroth-order optimization.” ICLR 2025.
>
> [3] Meng, Chuiyang, et al. “FLoRG: Federated Fine-tuning with Low-rank Gram Matrices and Procrustes Alignment.” ICLR 2026.
>
> [4] Liu, Zechun, et al. “Mobilellm: Optimizing sub-billion parameter language models for on-device use cases.” ICML 2024.

---

> > ### Author Rebuttal · Reviewer_dssz · 2026-04-02
> >
> > Thank you for the rebuttal.
> >
> > The claimed novelty of this work lies in a narrow, scenario-specific combination: namely, (1) a federated/split learning setting, (2) BP-free clients, and (3) server-side first-order (FO/BP) updates used to mitigate the slow convergence of pure zeroth-order methods. That said, the individual ingredients themselves are not new. First, hybrid FO/ZO federated optimization is not introduced here for the first time, as shown by VFL-CZOFO [1]. Second, client-BP-free federated learning is also not new in itself, as shown by BAFFLE [2]. Third, split/SFL with BP-free clients already has a closely related precedent in MU-SplitFed [3]. Therefore, my current view is that the claimed novelty is scenario-specific: it appears more like a plain combination and refinement of several existing ideas within a particular SFL setting, rather than a genuinely new optimization paradigm.
> >
> > As for the experimental section, within the line of zeroth-order LLM works, it is already common practice to evaluate on generative tasks that involve longer generation lengths and are generally harder to train, as well as on larger-scale LLMs (>10B, or even 66B) [4,5]. In particular, Corollary 4.5 indicates that the convergence rate deteriorates as the client-side dimension d_c increases. This suggests that the method’s performance cannot be assumed to scale well to general LLM sizes. Therefore, I remain skeptical about the scalability of the proposed algorithm.
> >
> > Accordingly, I will retain my score.
> >
> > ***
> >
> > **References**
> >
> > [1] Wang G, Gu B, Zhang Q, et al. A unified solution for privacy and communication efficiency in vertical federated learning[J]. Advances in Neural Information Processing Systems, 2023, 36: 13480-13491.
> >
> > [2] Feng H, Pang T, Du C, et al. Baffle: A baseline of backpropagation-free federated learning[C]//European Conference on Computer Vision. Cham: Springer Nature Switzerland, 2024: 89-109.
> >
> > [3] Liang D, Zhang J, Chen E, et al. Towards Straggler-Resilient Split Federated Learning: An Unbalanced Update Approach[J]. arXiv preprint arXiv:2510.21155, 2025.
> >
> > [4] Malladi S, Gao T, Nichani E, et al. Fine-tuning language models with just forward passes[J]. Advances in Neural Information Processing Systems, 2023, 36: 53038-53075.
> >
> > [5] Liu H, Wen R, Nair S, et al. Ecolora: Communication-efficient federated fine-tuning of large language models[C]//Proceedings of the 2025 Conference on Empirical Methods in Natural Language Processing. 2025: 20743-20757.

---

> > > ### Author Response · Authors · 2026-04-04
> > >
> > > Thank you for the continued discussions and your time.
> > >
> > > We want to highlight a fundamental principle: **the stacking of technologies does not result in the stacking of their benefits.** If BP (for fast convergence) is naively combined with ZO (for memory efficiency) in SFL, it does not automatically yield the "best of both worlds." In fact, it results in a significant extra communication costs. This bottleneck is explicitly acknowledged in **VFL-CZOFO**: *"The clients cannot calculate $\delta_{m,i}^j$ by themselves because the loss function $f_i(\cdot)$ is held by the server"*.  This coupling forces multiple (i.e., $P$-times) extra uplink transmissions of perturbed activations, causing massive communication overheads (as seen in MU-SplitFed).
> > >
> > > To solve this, we use **Lagrangian decomposition** to mathematically decouple the client and server objectives. This mathematical decoupling is the core reason why HO-SFL can achieve **all three benefits simultaneously**: FO-level performance, BP-free client memory, and dimension-free aggregation. Without this Lagrangian decomposition, relying solely on prior works cannot achieve these three advantages. We kindly note that applying this Lagrangian perspective in SFL **has not been introduced** in any papers discussed.
> > >
> > > Here is how HO-SFL fundamentally differs from the mentioned works:
> > >
> > > - **VFL-CZOFO:** It uses ZO only on one layer (i.e., the split layer) for **privacy protection**, while the rest of the client **still requires BP**. Moreover, it is a Vertical FL method, which is a completely different architecture where clients hold different features of the same sample.
> > > - **BAFFLE:** This is a pure ZO method. It suffers from severe convergence degradation and requires an impractical $P=500$ perturbations just to approach BP baselines.
> > > - **MU-SplitFed:** This is a pure ZO method for SFL. Since it lacks our Lagrangian decoupling, it cannot achieve client-local ZO evaluation, and its theoretical convergence rate depends on the dimension of the entire model. Crucially, as shown in our paper's Figure 3 and Figure 4a, its communication cost is massive and its convergence is slow compared with HO-SFL.
> > >
> > > **Regarding Scalability and Generative Tasks:** We scaled our client-side parameters by nearly 10x (from 125M to 1B) without any convergence degradation. While reviewer mentioned works like MeZO
> > > (which target >10B models), MeZO is a centralized algorithm. HO-SFL targets **edge devices**, where fitting a 10B+ model is physically impossible due to hard memory limits.
> > >
> > > Nevertheless, to address reviewer's concern for larger models and generative tasks, we have conducted new experiments fine-tuning **Qwen3-8B** and **LLaMA-3.2-3B** on the generative SQuAD dataset. The reported metric is the F1 score:
> > >
> > > |           | **OPT-125M** | **LLaMA-3.2-1B** | **LLaMA-3.2-3B** | **Qwen3-8B** |
> > > | --------- | ------------ | ---------------- | ---------------- | ------------ |
> > > | SplitLoRA | 0.5985       | 0.8804           | 0.9271           | 0.9413       |
> > > | HO-SFL    | 0.5744       | 0.8687           | 0.9238           | 0.9389       |
> > >
> > > Scaling the parameter size by a factor of 64x from the smallest model to the largest, as shown, HO-SFL continues to match the FO baseline effortlessly.
> > >
> > > Thanks again for your time and constructive feedback. We hope our explanations and new experiments successfully address your concerns and highlight the unique value of our work.

---

### Official Review · Reviewer_9MPT · 2026-03-24

**Soundness:** 3
**Presentation:** 4
**Significance:** 4
**Originality:** 3
**Overall Recommendation:** 4
**Confidence:** 3

**Summary:**

The paper deals with the problem of fine-tuning large models on resource constrained edge devices. It contends that backpropagation is infeasible on edge devices due to memory constraints, while zero order optimization is too slow and degrades statistical performance. It proposes HO-SFL (Hybrid Order Split federated learning) which is a reformulation of standard Split Federated Learning via variable lifting using Lagrangian multiplier technique. This reformulation enables a decoupling which lets the server undertake first order backpropagation updates whereas the clients are free to undertake zeroth order updates on their respective subnetworks. As a result, with model dimension-free aggregation from clients, the server can reconstruct a global gradient estimate that can support much faster convergence than a zeroth order method. This is demonstrated both theoretically and empirically. Theoretical convergence analysis also characterizes dependence on client-side parameter dimension rather than full model dimension. Experiments on CIFAR-10/100 and GLUE benchmarks with vision and language models demonstrate near-inference-level client memory consumption and competitive accuracy versus first-order Split FL baselines.

**Compliance With Llm Reviewing Policy:**

Affirmed.

**Final Justification:**

The authors addressed my main concerns. I have increased the score on presentation, but will keep my overall score unchanged.

**Key Questions For Authors:**

(Q1) There is no dual-update employed for the Lagrangian multiplier. Why not? Is it unnecessary?

(Q2) Algorithm 1, Line 10 - Impact of this transmission on privacy?

(Q3) Remark 4.6 - It seems to me like HO-SFL is gaining something without loosing anything. Could you explain?

(Q4) How should $P$ be set for models of different sizes?

**Limitations:**

The central server receives activation vectors and class labels from all clients. These can encode private information and this risk should be acknowledged.

**Strengths And Weaknesses:**

Strengths -

(S1) The theoretical convergence analysis formulation gives a non-trivially better result of convergence speed dependence on client side dimension post architectural split (rather than on full model dimension). Experiments are statistically well-supported and the memory footprint result is compellingat 4.14 GB client memory for LLaMA-3.2-1B versus 8.78 GB for standard Split FL, approaching the inference baseline of 4.01 GB.

(S2) The paper is well-written and the different parts of HOSFL are well explained with appropriate formalisms and intuitions.

(S3) The contribution is of high practical importance because of the set of desirable properties displayed by HO-SFL. Besides first order convergence rate behavior with zeroth order (a.k.a. inference baseline) memory consumption, communication overhead and latency stay minimal. This work will likely seed further research in the area.

(S4) While variable lifting with Lagrangian has a long history in optimization literature, its application to Split Federated Learning and the implications it brings seem novel.

Weaknesses -

(W1) There is closely related work which should be cited and against which the novelty of HO-SFL should be contextualized - (a) Hybrid Decentralized Optimization: Leveraging Both First- and Zeroth-Order Optimizers for Faster Convergence (AAAI 2025, https://ojs.aaai.org/index.php/AAAI/article/view/34290) has several elements of HO-SFL, (b) Zeroth-Order Fine-Tuning of LLMs in Random Subspaces (https://arxiv.org/abs/2410.08989v3) and HiSo: Efficient Federated Zeroth-Order Optimization via Hessian-Informed Acceleration and Scalar-Only Communication (https://neurips.cc/virtual/2025/loc/san-diego/124324) are faster ZO methods.

---

> ### Author Rebuttal · Authors · 2026-03-31
>
> Thank you for the constructive feedback. We are encouraged that you recognized the importance and novelty of our work. Below, we address your specific concerns.
>
> > W1: Missing Citations and Contextualization
>
> Thank you for highlighting these relevant works; we will cite and discuss them in our revision.
>
> - **Regarding [1]:** This work combines FO and ZO at the **node level** in a decentralized setting (some clients use FO, others ZO). In contrast, HO-SFL hybridizes at the **model split level**: *every* client uses ZO for their sub-model, while the server uses FO.
> - **Regarding [2] & [3]:** These papers propose advanced ZO estimators ( layer-wise low-rank perturbations and Hessian-informed acceleration). Our current implementation uses a standard Gaussian ZO estimator to prove the viability of decoupling FO and ZO. These advanced estimators are **orthogonal** to our framework, and integrating them to further accelerate HO-SFL is a promising direction.
>
> > Q1: Dual-Update for the Lagrangian Multiplier
>
> Iterative dual updates (like those in ADMM) are unnecessary here because the multiplier has an **analytical solution**. As shown in Section 3.2 (Eq. 10 and 11), enforcing the stationarity condition with respect to the activation $z$ directly yields $\lambda^t = \nabla_z l(f_s(z; \theta_s^t), y)$. The server’s standard BP computes this $\lambda^t$ automatically, bypassing the need for an iterative update step.
>
> > Q2 & Limitations: Privacy Impact of Transmitting Activations/Labels
>
> We acknowledge this risk and will add a “Privacy and Security Considerations” paragraph to the revised manuscript.
>
> - Transmitting activations is the fundamental mechanism of **all** SFL frameworks; this vulnerability is inherited from SFL, rather than introduced by HO-SFL.
> - Because our contribution changes the optimization method rather than the data flow, HO-SFL is **compatible** with standard privacy-enhancing defenses, such as applying Differential Privacy noise to the activations [4].
>
> > Q3: Gaining something without losing anything?
>
> While HO-SFL mitigates the dimensionality curse (Remark 4.6), we do sacrifice abundant or “hideable” resources to save strictly constrained ones:
>
> - **Trading Extra FPs for Memory:** In HO-SFL, clients perform $P$ extra ZO forward passes (FPs). We trade these extra FPs for critical memory savings (reducing the LLM footprint from 8.78 GB to 4.14 GB). These extra FPs “look” free because they can be executed within unavoidable transmission and server idle time (Section 3.4).
> - **Trading Clients’ Gradient Exactness for Communication:** By replacing client-side BP with ZO, we lose exact FO gradients, introducing approximation bias. However, we gain massive communication savings via dimension-free scalar aggregation.
>
> > Q4: Setting $P$ for Different Model Sizes
>
> We recommend a principled selection of $P$ based on the **model-specific latency-hiding bound**. While our ablations (Figure 8a) demonstrate that increasing $P$ monotonically improves accuracy, the marginal gains diminish gradually. Therefore, rather than sacrificing training speed for little returns, we recommend setting $P$ to the maximum value that fits within the system’s idle time. Crucially, this free budget scales **inversely with model size**. To demonstrate this better, we added a runtime simulation(https://anonymous.4open.science/r/HO-SFL-Figures/runtime_comparison.png) under the same settings as Appendix C.1. As illustrated in the figure, the end-to-end training time remains flat up to $P \approx 15$ for the smaller OPT-125M, but hits the latency-hiding limit at $P \approx 5$ for the larger LLaMA-3.2-1B. Thus, $P$ can simply be maximized up to the threshold where the runtime curve begins to bend upwards, ensuring better gradient estimation without any temporal overhead.
>
> Thank you again for your thorough review and valuable insights. We hope our rebuttal resolves your questions.
>
> ------
>
> > References
>
> [1] Talaei, Shayan, et al. “Hybrid Decentralized Optimization: Leveraging Both First-and Zeroth-Order Optimizers for Faster Convergence.” AAAI 2025.
>
> [2] Yu, Ziming, et al. “Zeroth-Order Fine-Tuning of LLMs in Random Subspaces.” ICCV2025.
>
> [3] Li, Zhe, et al. “HiSo: Efficient Federated Zeroth-Order Optimization via Hessian-Informed Acceleration and Scalar-Only Communication.” OPT 2025.
>
> [4] Wu, Maoqiang, et al. “Split learning with differential privacy for integrated terrestrial and non-terrestrial networks.” IEEE Wireless Communications 2023.

---

> > ### Author Rebuttal · Reviewer_9MPT · 2026-04-06
> >
> > I am satisfied with the authors' responses.

---

### Decision · Program_Chairs · 2026-04-30

**Decision:**

Accept (regular)

**Comment:**

This paper addresses an important problem in edge-oriented split/federated fine-tuning: standard client-side backpropagation is too memory-intensive, while pure zeroth-order optimization is often too slow. The proposed HO-SFL framework combines server-side first-order updates with client-side zeroth-order updates through a Lagrangian reformulation, aiming to obtain the benefits of both.

The reviewers agreed that the paper is practically motivated, clearly written, and technically solid overall. The main strengths are the useful hybrid design, the theoretical analysis showing that the zeroth-order penalty depends on the client-side subspace rather than the full model dimension, and the empirical results showing substantial client-memory and communication savings while remaining competitive with first-order baselines.

The main point of disagreement concerns novelty and experimental scope. One reviewer viewed the work primarily as a scenario-specific integration of existing ideas rather than a fundamentally new optimization paradigm, and also questioned whether the original experiments were broad enough. I agree that the novelty should be framed carefully: the paper does not introduce FO/ZO hybridization, split learning, or zeroth-order optimization individually. However, I am persuaded that the paper makes a sufficiently nontrivial contribution by formulating these ingredients in a way that achieves a meaningful joint tradeoff between memory, communication, and convergence in the split-learning setting.

The rebuttal and discussion strengthened the paper. In particular, the authors provided additional clarification and experiments addressing concerns about generative tasks, larger models, non-IID language settings, and sensitivity to perturbation-related hyperparameters. I have taken these responses into account, including the authors’ concern that one negative review did not fully reflect the updated evidence.

Overall, I find this to be a solid and practically relevant contribution with sufficient technical merit for ICML. For the final version, I encourage the authors to better position the work relative to prior FO/ZO and BP-free FL/SFL methods, and to more clearly discuss runtime tradeoffs, privacy considerations, and the regimes where HO-SFL is preferable to alternative approaches.